# De novo genome assembly depicts the immune genomic characteristics of cattle

Ting-Ting Li ®[1,6], Tian Xia ®[1,6], Jia-Qi Wu ®[1,6], Hao Hong[1], Zhao-Lin Sun[2], Ming Wang[3,4], Fang-Rong Ding[3], Jing Wang ®[2], Shuai Jiang[1], Jin Li[1], Jie Pan[1], Guang Yang[2], Jian-Nan Feng[2], Yun-Ping Dai[3], Xue-Min Zhang ®[1,5], Tao Zhou ®[1] ✉ & Tao Li ®[1,5] ✉

Immunogenomic loci remain poorly understood because of their genetic complexity and size. Here, we report the de novo assembly of a cattle genome and provide a detailed annotation of the immunogenomic loci. The assembled genome contains 143 contigs (N50 ~ 74.0 Mb). In contrast to the current reference genome (ARS-UCD1.2), 156 gaps are closed and 467 scaffolds are located in our assembly. Importantly, the immunogenomic regions, including three immunoglobulin (IG) loci, four T-cell receptor (TR) loci, and the major histocompatibility complex (MHC) locus, are seamlessly assembled and precisely annotated. With the characterization of 258 IG genes and 657 TR genes distributed across seven genomic loci, we present a detailed depiction of immune gene diversity in cattle. Moreover, the MHC gene structures are integrally revealed with properly phased haplotypes. Together, our work describes a more complete cattle genome, and provides a comprehensive view of its complex immune-genome.

The immune system contains the largest source of genetic variation. Its prodigious diversity and complexity ensure that the host can precisely distinguish non-self from self and effectively respond to the persistent, but unpredictable, environmental challenges[1]. At the DNA level, the immunogenomic loci of T cells and B cells represent the typical examples of the genetic variations[2,3]. During the maturation of B/T cells, a process known as V(D)J recombination occurs. This process combines randomly selected individual segment from each of the preexisting variable (V), diversity (D), and joining (J) gene clusters and give rise to the tremendous diversity of IG/TR on mature B/T cells[4]. Each B or T cell, as characterized by a uniquely expressed IG or TR gene, can response to a specific antigen. Together, the IG and TR genes encode a major part of the immune repertoire[5]. Another example is the MHC gene locus, which consists of many genes that are involved in the immune defense systems and exhibit the highest diversity among the population[6]. Because of the structural complexity of these immunogenomic loci, a comprehensive description of these regions remains a challenge. The complete assembly and annotation of the immunogenomic loci will provide fundamental and accurate descriptive data for immunological studies. Recently, using nanopore sequencing technology, the human MHC gene locus was completely assembled and phased with ultra-long reads[7].

The average cost of de novo assembly of a genome has markedly decreased because of the availability of next generation sequencing (NGS) technologies, such as the Illumina platform[8]. More importantly, the third-generation sequencing technologies, which generate long reads that exceed dozens of kilobases, have resulted in a paradigm shift to enable whole-genome assembly, not only for experimental methods, but also for algorithms[9]. Pacific Biosciences (PacBio) "single-molecule real time" methods can generate ~10 Kb of long HiFi reads

[1]Nanhu Laboratory, National Center of Biomedical Analysis, Beijing 100850, China. [2]State Key Laboratory of Toxicology and Medical Countermeasures, Beijing Institute of Pharmacology and Toxicology, Beijing 100850, China. [3]State Key Laboratories for Agrobiotechnology, College of Biological Sciences, China Agricultural University, No.2 Yuanmingyuan Xilu, Beijing 100193, China. [4]College of Animal Science and Technology, China Agricultural University, No.2 Yuanmingyuan Xilu, Beijing 100193, China. [5]School of Basic Medical Sciences, Fudan University, Shanghai 200032, China. [6]These authors contributed equally: Ting-Ting Li, Tian Xia, Jia-Qi Wu. ✉e-mail: tzhou@ncba.ac.cn; tli@ncba.ac.cn

with 99% accuracy[10]. Oxford Nanopore Technologies (ONT) recently developed an ultra-long read method that produces reads with an average length of ~50 Kb, and the longest reads can reach hundreds of kilobases or even over mega-bases[7,11]. The incredible technical progress has resulted in an abundance of genome assemblies for animals, plants, and other organisms. For the human genome, the assembly of a centromere on the Y chromosome[12], telomere-to-telomere assembly of a specific chromosome[13,14], and a real gapless assembly of all 22 autosomes plus an X chromosome[11] were recently reported. These advances provide detailed data and a panoramic view of genomic diversity, particularly the immune-genome of human.

As one of the most important livestock, cattle have made important historical contributions and are continuing to contribute to our understanding of the basic and applied immunology[15–18]. Recent studies on the gene structure of bovine immunogenomic loci have significantly advanced our understanding of the bovine immune-genome, such as MHC[19,20], IG[21,22] and TR[23–26]. Interestingly, the long third heavy chain complementary determining regions (CDRH3) in cattle are capable of rapidly generating broad-neutralizing antibodies against human immunodeficiency virus (HIV)[27]. Furthermore, the profiling of CDRH3[28], γδ TCR[29], and MHC diversities[30–32] revealed the specific and large expansion of the bovine immune repertoire and the genomic basis and essence of cattle immunity[33,34]. However, the incomplete understanding of the cattle genome has limited the in-depth studies on the complex immunogenomic loci of this important animal. Although several genome assemblies, including the current reference cattle genome, ARS-UCD1.2, were reported previously[35–37], the genome continuity and completeness of these assemblies are still limited. A high-quality reference genome is necessary to facilitate research on cattle immunity.

In this study, we report the assembly of the cattle genome using a combination of several advanced sequencing methods, in particular, the ONT ultra-long read sequencing technology. Our results significantly surpass the continuity and accuracy of ARS-UCD1.2, and generate a gapless assembly with refined annotation of the immunogenomic loci, including IG, TR and MHC loci.

## Results

### De novo assembly of a cattle genome
To assemble a more complete and accurate genome version of the cattle genome, we performed whole-genome sequencing of female cattle embryonic fibroblasts using ONT ultra-long read sequence technology, PacBio circular consensus sequencing (HiFi) and Illumina NGS, combined with Hi-C and BioNano optical mapping (Supplementary Data 1). A total of 499.0 Gb of ultra-long reads were generated with an N50 length of 70.4 Kb, and the longest read was 872.5 Kb. The ONT data was superior in read-length compared with that of the PacBio and Illumina methods (Supplementary Fig. 1a–d).

Using NextDenovo, we performed a de novo assembly of the ultra-long reads (Supplementary Fig. 1e). The assembly consists of only 143 primary contigs with an N50 contig size of 74.01 Mb and a total length of 2.68 Gb (Supplementary Data 2). The contigs were polished and corrected with the PacBio HiFi reads and Illumina reads, anchored with BioNano optical maps and Hi-C data (Supplementary Fig. 1e) into a final genome assembly with excellent continuity (Supplementary Fig. 1f). We designated our assembly as NCBA_BosT1.0 (Fig. 1a, and Supplementary Data 3). The N50 scaffold size was 74.72 Mb and the final genome size was 2.71 Gb. To determine the completeness and accuracy of the NCBA_BosT1.0, we aligned all the reads back to this newly assembled genome, and 99% of the whole genome exhibited a minimum coverage of 53× by ultra-long reads, 4× by PacBio HiFi reads and 64× by Illumina reads (Fig. 1b, Supplementary Fig. 1g, and Supplementary Data 4). The ultra-long reads showed significant low-bias with respect to the GC content compared with that of the Illumina reads and HiFi reads, particularly in GC-poor regions (Supplementary Fig. 1h). We

constructed 3D genome heatmaps of the NCBA_BosT1.0 assembly using the in situ Hi-C method. All of the chromosomes exhibited clear intra-chromosomal diagonal signals with no significant inter-chromosomal signals (Supplementary Fig. 2a), which indicated the completeness and continuity of the new assembly, as well as the low mis-assembly rate of the structural variations. We further evaluated the correctness of NCBA_BosT1.0. To do so, Illumina reads were mapped onto the assembly and homozygous variations were called. We identified 10,813 homo SNPs (error rate 0.0004%) and 10,738 homo Indels (error rate 0.0008%) with a minimum coverage of 5. This suggests an average consensus single base accuracy of Phred Q51 (Supplementary Fig. 2b). Thus, our data indicate that the ultra-long read sequencing provides the outstanding performance for genome assembly.

### Gap filling and genome annotation
The NCBA_BosT1.0 assembly demonstrated superior sequence integrity compared with the current cattle reference genome, ARS-UCD1.2, as well as other cattle genomes (Supplementary Fig. 2c). Of the 30 chromosomes, 13 chromosomes were packaged by one single contig (Fig. 1a, and Supplementary Data 5). Because all autosomes of the cattle genome are acrocentric, no telomere with p-arms was successfully assembled due to the satDNA arrays and segmental duplications around the centromeres. The q-arms of seven chromosomes were ended with a minimum of 15-Kb (TTAGGG)$_n$ telomere repeats, of which five chromosomes (chromosomes 17, 20, 22, 26, and 28) represented gapless centromere-to-telomere assemblies (Fig. 1a). The gap-remaining regions were primarily localized to the acrocentric regions of p-arms and the chromosome X (Supplementary Data 5). Next, we assessed whether the remaining gaps in ARS-UCD1.2 could be filled using our assembly. The ARS-UCD1.2 contains 30 chromosomes and 2180 unplaced scaffolds. There are 386 gaps denoted as Ns and 315 gaps of which are localized on chromosomes. With our assembly, 156 gaps on chromosomes were filled (Fig. 1a, b, and Supplementary Data 6). In addition, 420 scaffolds and 47 partial scaffolds, with a total length of 24.89 Mb, were properly inserted back into our new genome (Supplementary Fig. 3, and Supplementary Data 7). These scaffolds ranged from Kbs to Mbs and the largest scaffold was 4.3 Mb on chromosome X.

We further annotated the newly assembled genome. Transposable element repeats (TE) accounted for 46.53% of NCBA_BosT1.0 and the total ratio of repeat sequences was 47.30% (Fig. 1c, and Supplementary Data 8). A total of 20,288 genes were predicted with an average length of 40.4 Kb, which was consistent with bovine and other proximal species (Fig. 1d, Supplementary Fig. 4, and Supplementary Data 8). Based on a benchmarking universal single-copy ortholog (BUSCO) analysis, we found that 96.1% of the BUSCO genes were included in our assembly, indicating that NCBA_BosT1.0 is nearly complete (Fig. 1d). We also validated the expression of the predicted genes by RNA-Seq and confirmed the expression of 89.7% of these genes (Fig. 1d). Functional annotation of the genes using databases, including Kyoto Encyclopedia of Genes and Genomes (KEGG), Gene Ontology (GO), euKaryotic Orthologous Groups of protein (KOG), Non-Redundant Protein Database (NR), and Swiss-Prot, revealed both high coverage and intersection ratios (Supplementary Fig. 5, and Supplementary Data 8). These data demonstrate the reliability of the NCBA_BosT1.0 genome annotation.

### Seamless assembly and annotation of immunoglobulin gene loci
The assembly of the cattle genome with higher continuity and completeness enabled us to depict detailed gene structures of the complex immunogenomic loci. We primarily evaluated the IG, TR, and MHC gene clusters as well as natural killer (NK) cell receptors, which are localized on eight different chromosomes (Supplementary Fig. 6). The IG genes are located on three gene loci, immunoglobulin heavy chain (IGH), lambda chain (IGL) and kappa chain (IGK). All IG-related gene

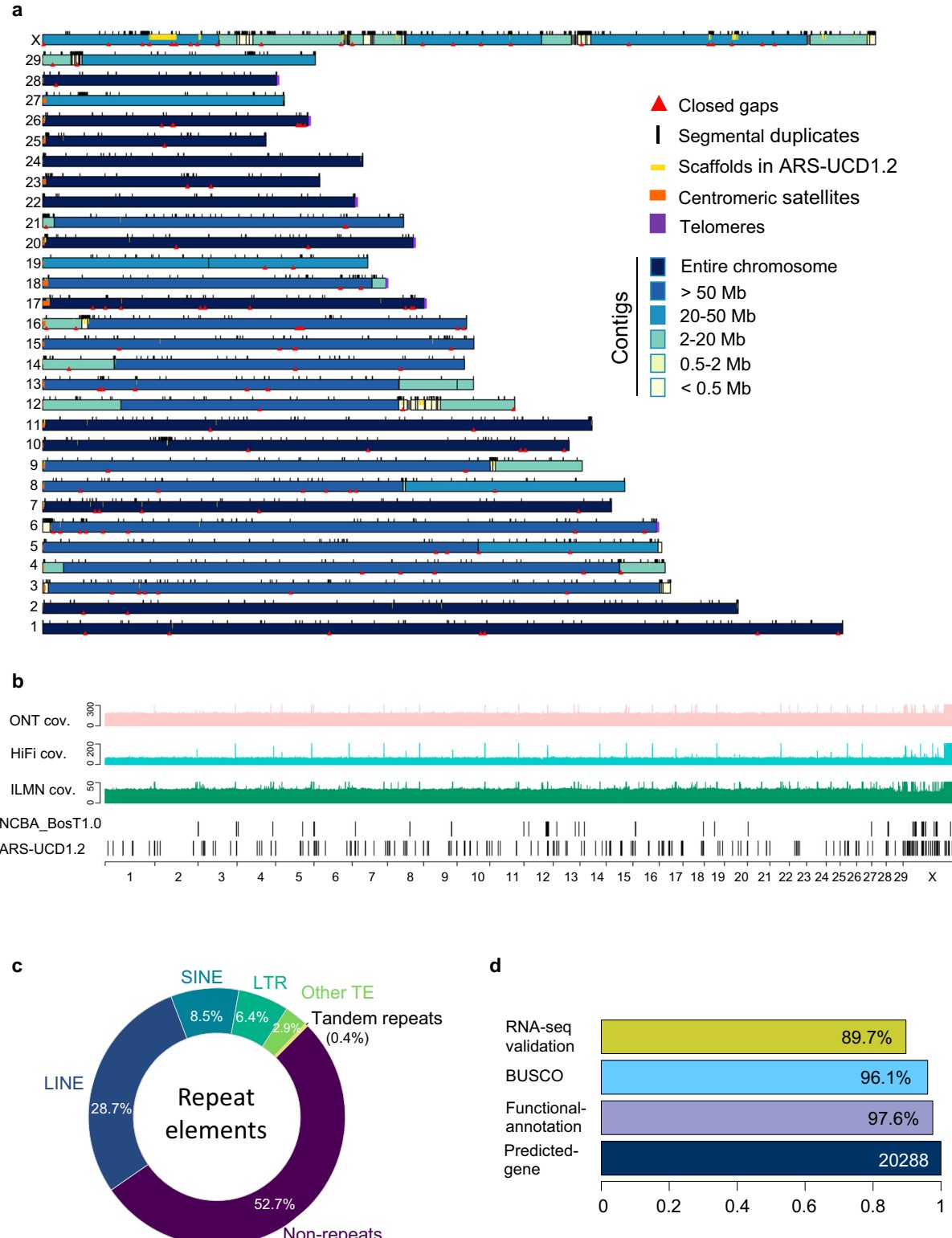

**Fig. 1 | A global picture of the de novo cattle genome assembly. a** Ideograph of cattle genome assembly NCBA_BosT1.0. Chromosomes composed of a single contig are in dark blue, satDNAs are in orange and telomeres are in purple. Closed gaps and properly placed scaffolds of ARS-UCD1.2 are depicted as red triangles and yellow strips. **b** Read coverage of cattle genome NCBA_BosT1.0. Lane 1–3: read coverages by ONT ultra-long reads, PacBio HiFi reads and Illumine NGS reads. Lane 4–5: remaining gaps in NCBA_BosT1.0 and ARS-UCD1.2. **c** Composition ratio of repeat elements in NCBA_BosT1.0. **d** Gene annotations of NCBA_BosT1.0. There were 20,288 genes predicted in total, and the ratio of genes with functional annotation, overlapped with BUSCO and validated with RNA-Seq were indicated.

loci were covered with gapless contigs and were well annotated in NCBA_BosT1.0 (Fig. 2, and Supplementary Fig. 6), whereas in ARS-UCD1.2, IGH is incorrectly assembled into three regions (Supplementary Fig. 7) and the IGL region contains six gaps (Fig. 2c). In line with previous study[38], We maintained the same gene name established in the IMGT database, and provisionally named the newly discovered genes (with asterisks). The detailed gene structure and functional annotations of the IG/TR loci were generated following IMGT criteria (Supplementary Fig. 8).

The IGH was 616.0 Kb in size and was localized to the end of the q-arm of chromosome 21 (Fig. 2a, b, and Supplementary Fig. 9a). In NCBA_BosT1.0, the IGH locus contained 48 IGHV genes (11 functional) belonging to 3 IGHV subgroups as well as 17 IGHD (all functional), 12 IGHJ (3 functional) and 10 IGHC (8 functional) genes (Fig. 2b). A previous study assembled the IGH locus by sequencing seven BAC clones and generated an IGH gene structure containing three tandem [IGHDP-IGHV3-(IGHDv)$_{5/6}$] repeats[21]. In contrast, we observed only two tandem repeats [IGHDP-IGHV3-(IGHDv)$_{5/6}$] in the IGH locus of NCBA_BosT1.0, and the repeat regions as well as the adjacent gene loci were fully covered with multiple ultra-long reads greater than 100 Kb (Supplementary Fig. 9b, and Supplementary Fig. 10). These data ensured the accuracy and reliability of our sequence assembly. In addition to the above data, two extra IGHV genes in the V region were identified (Fig. 2b). Thus, our results suggest that the IGH locus in NCBA_BosT1.0 is organized as: (IGHV)$_{46}$-(IGHDv)$_5$-(IGHJ)$_6$-IGHM1-[IGHDP-IGHV3-(IGHDv)$_{5/6}$]$_2$-(IGHJ)$_6$-IGHM2-IGHD-IGHG3-IGHG1-IGHG2-IGHE-IGHA (Fig. 2a, b).

The IGL locus spanned 643.9 Kb on the reverse strand of the q-arm on chromosome 17. By filling in the remaining six gaps in the IGL locus of ARS-UCD1.2, 125 IGLV genes (37 functional) were annotated, of which 51 IGLV genes were newly identified (Fig. 2c, and Supplementary Fig. 11a). Of note, we characterized six repeats of IGLJ-IGLC clusters, whereas in ARS-UCD1.2, there are nine repeats. (Fig. 2c, and Supplementary Fig. 11a, b). The whole IGL genome locus was covered with an average depth of 13 using ultra-long reads greater than 100 Kb and four ultra-long reads spanning over the entire IGLC region (Supplementary Fig. 12a), which yielded reliable evidence for the genome assembly and annotation of the IGL locus. These results indicate that the IGL genes are organized as: (IGLV)$_{125}$-(IGLJ-IGLC)$_6$.

The IGK is the smallest gene locus of the IGs and spanned 214.3 Kb between 47.2 Mb and 47.4 Mb on chromosome 11 (Fig. 2d, and Supplementary Fig. 12b). IGK consisted of 28 V genes (7 functional), 5J genes (1 functional) and 1C gene in NCBA_BosT1.0, and 3 new V genes were identified compared with ARS-UCD1.2 (Fig. 2d, and Supplementary Fig. 12b). Our data suggest that the IGK locus is organized as (IGKV)$_{28}$-(IGKJ)$_5$-IGKC.

### Gapless assembly and annotation of the T cell receptor gene loci

The T-cell receptors for cattle are composed of four gene subgroups, TRA, TRB, TRD and TRG (Supplementary Fig. 6). The TRA/TRD region covered 3.3 Mb on the reverse strand of chromosome 10, consisted of over 400 V genes[23], and was the most complex immunogenomic locus in cattle (Fig. 3a, and Supplementary Fig. 13a). The entire TRD resides within the TRA region (Fig. 3a, b). Interestingly, the V region of TRA/TRD spanned over 3 Mb and accounted for >90% of the total TRA/TRD region. In NCBA_BosT1.0, we annotated 305 TRAV genes (148 functional) and 64 TRDV genes (48 functional), whereas in ARS-UCD1.2, only 183 TRAV genes (85 functional) and 39 TRDV genes (31 functional) are identified. In line with ARS-UCD1.2, the D-J-C cluster consisted of 60 TRAJ genes, 1 TRAC gene, 9 TRDD genes, 4 TRDJ genes and 1 TRDC gene (Fig. 3b). Therefore, our data suggest that the TRA(D) genomic structures are organized as [TRA(D)V]$_{368}$-(TRDD)$_9$-(TRDJ)$_4$-TRDC-TRDV3-(TRAJ)$_{60}$-TRAC.

The TRB locus spanned 667.3 Kb between 105.4 Mb and 106.2 Mb on chromosome 4, and consisted of 153 TRBV genes (87 functional),

3 TRBD genes (all functional), 19 TRBJ genes (15 functional), and 3 functional TRBC genes (Fig. 4a, b). These data closed the previously remaining two gaps in ARS-UCD1.2 (Fig. 4a, and Supplementary Fig. 13b). The TRBD, TRBJ, and TRBC genes were organized into three tandem D-J-C cassettes, followed by one functional TRBV gene (TRBV30) in an inverted orientation (Fig. 4b). Thus, the TRB genomic structures are organized as: (TRBV)$_{152}$-[TRBD-(TRBJ)$_{6/7}$-TRBC]$_3$-TRBV30.

The TRG genes are localized at two separate loci on chromosome 4. They were on different strands and were 30 Mb apart from one another (Fig. 4c, and Supplementary Fig. 13c). TRG1 spanned 229.3 Kb and consisted of four tandem V-J-C cassettes, whereas TRG2 spanned 106.0 Kb and consisted of three tandem V-J-C cassettes (Fig. 4c). In total, TRG consisted of 18 TRGV (17 functional), 10 TRGJ (8 functional), and 7 TRGC (all functional) genes. The TRG genes are organized as [(TRGV)$_{1/5/7}$-(TRGJ)$_{1/2}$-TRGC]$_4$ for TRG1 and [(TRGV)$_{1/2}$-(TRGJ)$_{1/2}$-TRGC]$_3$ for TRG2.

In summary, the NCBA_BosT1.0 cattle genome contains 915 IG and TR genes (258 IG and 657 TR) that are localized at seven major loci and distributed in 741 V, 29 D, 116 J and 29 C genes (Table 1). The elaborate annotations of NCBA_BosT1.0 markedly extended our understanding of the gene sequence diversity, particularly with respect to the TR genes (Supplementary Data 9). Our results provide important data that reveal the dramatic diversity of the cattle immune repertoire.

### Phylogenetic analysis of the V genes

We performed a phylogenetic analysis of V genes across species. Gene diversity statistics of species with detailed IG/TR annotations revealed that NCBA_BosT1.0 has the highest IG/TR gene numbers among known species, particularly for the TR genes (Supplementary Data 9). Phylogenetic trees were constructed with all functional IG- and TR-V genes from human, mouse, and cattle (NCBA_BosT1.0). The results indicated that all of the V genes were well clustered according to their subgroups. Interestingly, TRAV genes were clustered into two separate groups (Supplementary Fig. 14). These results demonstrate the evolutionary conservation of IG and TR genes among species. The gene number analysis also reveals a significant deviation: both human and mouse had much more IG V genes than TR V genes, whereas cattle exhibited the opposite with threefold more TR V genes than IG V genes (Fig. 5a and Supplementary Data 9).

### Full-length IG/TR transcriptome profiling

To determine the preference and frequency of V-D-J segment usage during recombination, we profiled the immune repertoires of cattle using PacBio HiFi full-length transcriptome sequencing. Peripheral blood mononuclear cells (PBMC) were collected from four cattle and then the cDNA libraries were prepared and sequenced. We identified a total of 36,751 sequences, including 18,493 IGs and 18,258 TRs, and over 90% of these sequences are functional (Fig. 5b). Moreover, IGL was superior to IGK for cattle light chain usage (Fig. 5b), which is in accordance with a previous study[39]. We also analyzed the ultra-long CDRH3-containing sequences and found an exclusive selectivity of IGHV1−7 and IGHD8-2 (Fig. 5c), which was also consistent with previous studies[40,41]. The ultra-long CDRH3s exhibited the highest abundance in the IGH repertoires with IGHJ2−4 as the most dominant J segment (Fig. 5d).

By analyzing the V(D)J recombination profiles, we found that, impressively, cattle exhibited a significantly more intricate and diverse TR repertoire compared with the IG repertoire (Fig. 5d−j). For IGs, the V segments of light chains exhibited a relative even frequency of usage during recombination. In contrast, the heavy chains showed strong preference for V segment selection (Fig. 5d−f). Of the TRs, TRA showed the highest diversity according to the numbers of types of V-J recombination. Among the combinations, TRAV3-5-TRAJ10 and TRAV33-3-TRAJ10 were expressed at relatively higher levels (Fig. 5g). TRB exhibited the second highest complexity (Fig. 5h), thus making up the extraordinary TCRα:β heterodimer repertoire together with TRA.

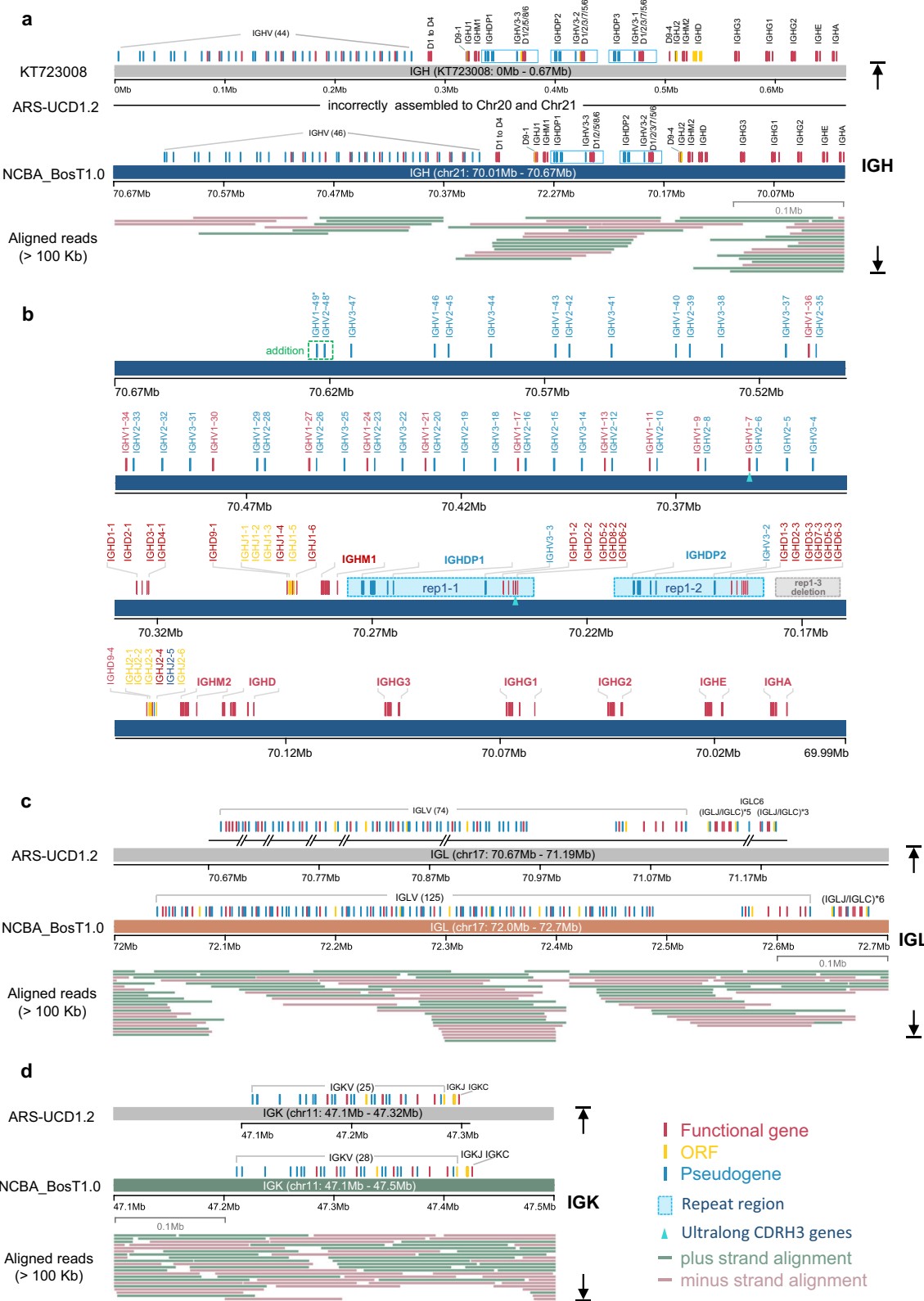

**Fig. 2 | The cattle immunoglobulin loci. a** Genomic organization of IGH locus in KT723008, ARS-UCD1.2 and NCBA_BosT1.0. Repeated regions were drawn as blue rectangles, and ONT ultra-long reads that longer than 100 Kb and the mapping to the genomic region were drawn. In newly assembled NCBA_BosT1.0, there is a deletion of tandem repeat [IGHDP-IGHV3-(IGHDv)5/6] in IGH locus. **b** Detailed diagram of IGH gene structure and annotation in NCBA_BosT1.0. Two repeated regions were labeled as blue rectangles (rep1-1 and rep1-2), and a deletion of repeat

region (rep1-3) was labeled as gray rectangle. IGHV1–7 and IGHD8-2 consists of the ultra-long CDRH3 were marked with green triangles. The newly identified IGHV genes in NCBA_BosT1.0 were assigned provisional names with asterisk. **c, d** Genomic organization of IGL and IGK loci in ARS-UCD1.2 and NCBA_BosT1.0. Genomic gaps in ARS-UCD1.2 were depicted beneath the genes according to their coordinates and ONT ultra-long reads that longer than 100 Kb and the mapping to the genomic region were drawn.

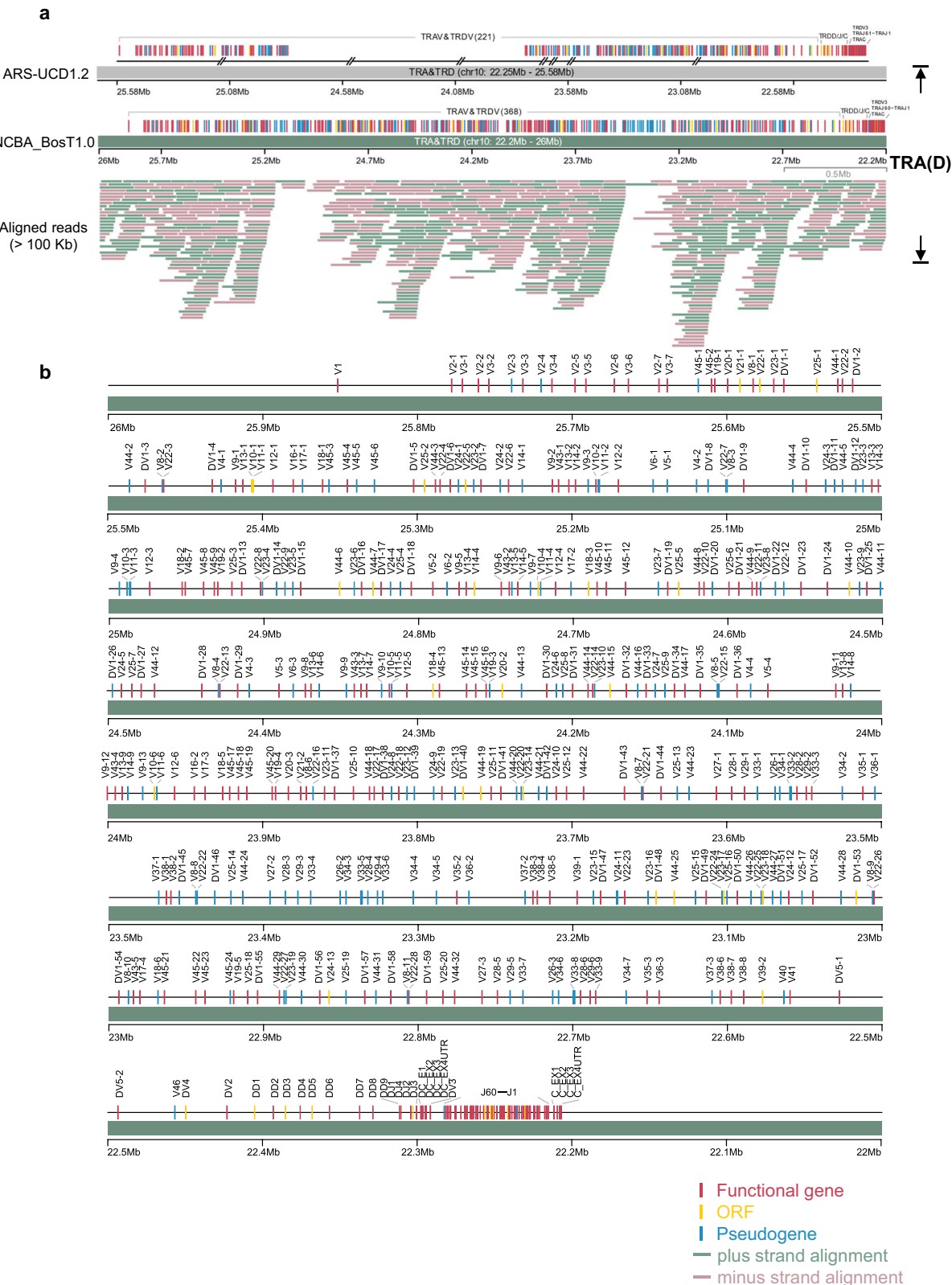

**Fig. 3 | The cattle TRA/TRD loci. a** General organization of TRA/TRD loci in ARS-UCD1.2 and NCBA_BosT1.0. Genomic gaps in ARS-UCD1.2 and NCBA_BosT1.0 were depicted beneath the genes according to their coordinates. ONT ultra-long reads that longer than 100 Kb and the mapping to the genomic region were drawn. **b** The detailed genetic map of TRA/TRD loci in NCBA_BosT1.0. Labels of TRD genes starts with "D" and all TRD genes reside within the TRA genomic region.

These data were consistent with that of a previous study[29]. Similarly, we also analyzed the V-(D)-J recombination profile of TRD and TRG (Fig. 5i, j). Taken together, our data reveal the distinctive patterns in the IG and TR immune repertoires of cattle.

**The complete assembly and annotation of the MHC gene locus**
MHC plays a crucial role in determining immune responsiveness and is known as the bovine leukocyte antigen (BoLA) in cattle. MHC genes primarily consist of two clusters, MHC class I and MHC class II[6].

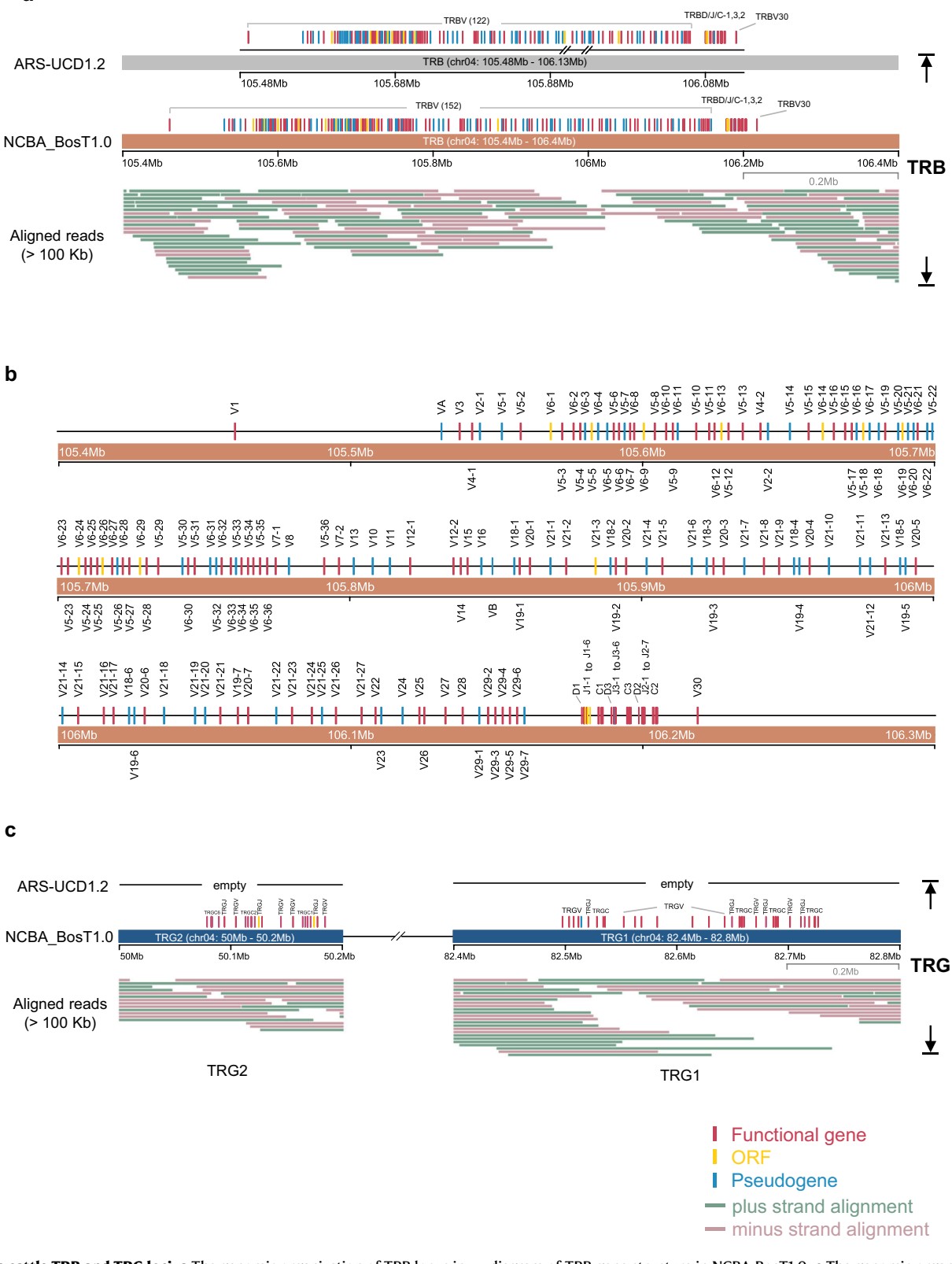

**Fig. 4 | The cattle TRB and TRG loci. a** The genomic organization of TRB locus in ARS-UCD1.2 and NCBA_BosT1.0. Genomic gaps in ARS-UCD1.2 were depicted beneath the genes according to their coordinates. ONT ultra-long reads that longer than 100 Kb and the mapping to the genomic region were drawn. **b** The detailed diagram of TRB gene structure in NCBA_BosT1.0. **c** The genomic organization of TRG loci in NCBA_BosT1.0. ONT ultra-long reads that longer than 100 Kb and the mapping to the genomic region were drawn.

**Table 1 | Gene numbers of each immune locus in NCBA_BosT1.0 assembly**

| Types | IGH | IGK | IGL | TRA | TRB | TRD | TRG1 | TRG2 | Sum |
|---|---|---|---|---|---|---|---|---|---|
| V (F/P/ORF) | 48 (11/37/0) | 28 (7/19/2) | 125 (37/80/8) | 305 (148/132/25) | 153 (87/55/11) | 64 (48/12/4) | 14 (13/1/0) | 4 (4/0/0) | 741 (355/336/50) |
| D (F/P/ORF) | 17 (17/0/0) | NA | NA | NA | 3 (3/0/0) | 9 (6/0/3) | NA | NA | 29 (26/0/3) |
| J (F/P/ORF) | 12 (3/1/8) | 5 (1/0/4) | 6 (4/0/2) | 60 (53/2/5) | 19 (15/1/3) | 4 (3/0/1) | 5 (5/0/0) | 5 (3/0/2) | 116 (87/4/25) |
| C (F/P/ORF) | 10 (8/2/0) | 1 (1/0/0) | 6 (3/3/0) | 1 (1/0/0) | 3 (3/0/0) | 1 (1/0/0) | 4 (4/0/0) | 3 (3/0/0) | 29 (24/5/0) |
| Sum (F/P/ORF) | 87 (39/40/8) | 34 (9/19/6) | 137 (44/83/10) | 366 (202/134/30) | 178 (108/56/14) | 78 (58/12/8) | 23 (22/1/0) | 12 (10/0/2) | |

Functional annotation for each gene of IGs and TRs was performed based on the IMGT criteria (Supplementary Fig. 8), and gene numbers were collected and listed in the table.

Haplotypes of MHC I have been sequenced and annotated using BAC libraries[19,20], and were further gap-filled in ARS-UCD1.2[35]. The cattle genome contains six classical MHC I genes (BoLA 1–6) with high polymorphism and ten non-classical MHC I genes (NC1–10) that exhibit limited polymorphism[20,42]. In NCBA_BosT1.0, the BoLA genetic region was covered by one single contig and ranged over 3.38 Mb on chromosome 23 (Fig. 6a, and Supplementary Fig. 15). By acquiring the seamless sequence of this gene locus, we were able to coordinately annotate the MHC genes. Classical MHC I genes were located at the 3′end of the entire BoLA region. The non-classical MHC I genes NC6–10 were adjacent to the classical MHC I genes and NC2–5 was located 600 Kb upstream away (Fig. 6b). For the MHC II genes, there were only DQ and DR gene pairs that were organized in adjacent sequential order (Fig. 6c), which was in line with a previous study[43]. In contrast, the human MHC II gene locus harbors an additional DP gene pair[7]. We also annotated other genes located within the BoLA region, and most of these genes are functionally related with the immune response, such as C2 and IL17 (Fig. 6b, c).

To better understand the gene organizations of the BoLA region, haplotypes were phased with ONT ultra-long reads combined with heterozygous SNPs, which were called using Illumina data and HiFi reads. The N50 length of haplotype 1 and haplotype 2 were 198.5 Kb and 200.6 Kb, respectively (Fig. 6d, e, and Supplementary Data 10). Of the 3.38 Mb BoLA region, 3.17 Mb were successfully assembled into haplotypes and both haplotypes exhibited high continuity (Supplementary Fig. 15b). The sequence of each haplotype showed delicate differences in gene structures for both MHC I and MHC II (Fig. 6f), demonstrating the polymorphism and polygeny of BoLA among individuals. For example, for the MHC II gene locus, there were two [DQB/DQA] gene pairs within haplotype 1, while only one [DQB/DQA] gene pair was identified in haplotype 2, and all the loci of MHC II genes were validated with PacBio full-length transcripts (Fig. 6f). Likewise, we showed that the classical MHC I genes varied in terms of both gene types and numbers between the two haplotypes (Supplementary Fig. 15a). In summary, the BoLA was assembled and phased over its full length in a diploid cattle genome. These data also highlight the power of ONT ultra-long technology in resolving the haplotypes of highly intricate gene clusters of large size.

NK cell receptors, which interact with MHC molecules, are encoded by gene families exhibiting high monomorphic genetic diversity[44]. The genes encoding NK cell receptors are clustered into two main gene complexes: the natural killer complex (NKC), which encodes C-type lectin-like molecules, and the leukocyte receptor complex (LRC) that encodes killer immunoglobulin-like receptors (KIRs)[44]. Both loci were seamlessly assembled in NCBA_BosT1.0 (Supplementary Fig. 16). In terms of NK gene complex, the global gene structures exhibit a high degree of similarity between NCBA_BosT1.0 and ARS-UCD1.2, however, a notable difference existed in the gene count of KLRC1 and its nearby homologs (Supplementary Fig. 16a). NCBA_BosT1.0 contained one KLRC1 gene and five nearby highly similar genes (KLRC1- [2–6]) whereas ARS-UCD1.2 contains one KLRC1 gene and two similar KLRC1 genes (LOC100847738 and LOC100336869). For the LRC gene complex, NCBA_BosT1.0 contained 17 KIR genes in total, including one

KIR2DL gene (2DL5A), two KIR2DS genes, eight KIR3DS genes and six KIR3DL genes. The gene numbers in this region were significantly more than that of ARS-UCD1.2 (Supplementary Fig. 16b).

Taken together, by assembling a cattle genome with increased continuity and completeness, we annotated and profiled the IG/TR genomic structures, as well as the MHC and NK cell receptor genomic loci from de novo. Our study provides a complete immunogenomic landscape for cattle.

## Discussion

Sequencing-technology progress, especially the ONT ultra-long sequencing method, has greatly promoted the complete assembly of genome. For example, the CHM13, a telomere-to-telomere human genome assembly that contains gapless sequences for all chromosomes except Y was reported recently[11]. As the immune system possesses the biggest source of genetic variation, depicting the immunogenomic loci was almost impossible before, let alone population genomic diversity studies of the immune system. A completely assembled genome, or at least completely assembled immunogenomic loci, can be the cornerstone for the in-depth understanding of the immune system of given species. In this study, by taking advantage of the ONT ultra-long sequencing method, we report a cattle genome assembly that exhibit remarkable improvement over existing assemblies. Importantly, by assembling the complete DNA sequences of IG, TR, and MHC loci, as well as NK receptor loci, we delineate the complex genomic structures of the immune-genome of cattle, and provide fundamental immunogenomic data for further studies.

Our work affords a more precise roadmap of cattle immune genome. We characterized and rigorously demonstrated the tandem repeat patterns within the IGH and IGL regions, and filled 15 gaps within the IG and TR loci. 741 V genes were annotated in NCBA_BosT1.0, whereas only 524 V genes are currently collected in the IMGT database (Table 1, and Supplementary Data 9). The cattle MHC region was also seamlessly packaged and properly phased, which is the second intact MHC assembly besides human to our knowledge[7].

Cattle exhibits distinct characteristics in both IG and TR immune repertoire. For IGH genes, there were only 48 V genes, which are significantly less than that of human and mouse. The relatively low diversity of IG in cattle is likely compensated by the ultra-long CDRH3s that are found in ~10% of the immunoglobulins[45]. The ultra-long CDRH3s allow cattle antibodies to bind a wider range of antigens and play a key role in neutralizing HIV infections[27,40,41]. These cattle ultra-long CDRH3s almost exclusively use the same V gene segment (IGHV1–7) that contains an eight-nucleotide duplication "TACTACTG" at its 3′ end, and the same D gene segment (IGHD8-2) known as the longest D gene[41]. Both IGHV1–7 and IGHD8-2 gene loci were clearly depicted in NCBA_BosT1.0. Our study provides the complete DNA sequence of this region, and based on these data, the regulation of V-to-DJ rearrangements between these two loci can be further investigated.

Cattle possess the highest TR gene diversities among all species annotated with detailed V(D)J gene structures (Supplementary Data 9). We identified 540 TR V genes in NCBA_BosT1.0, which was three times

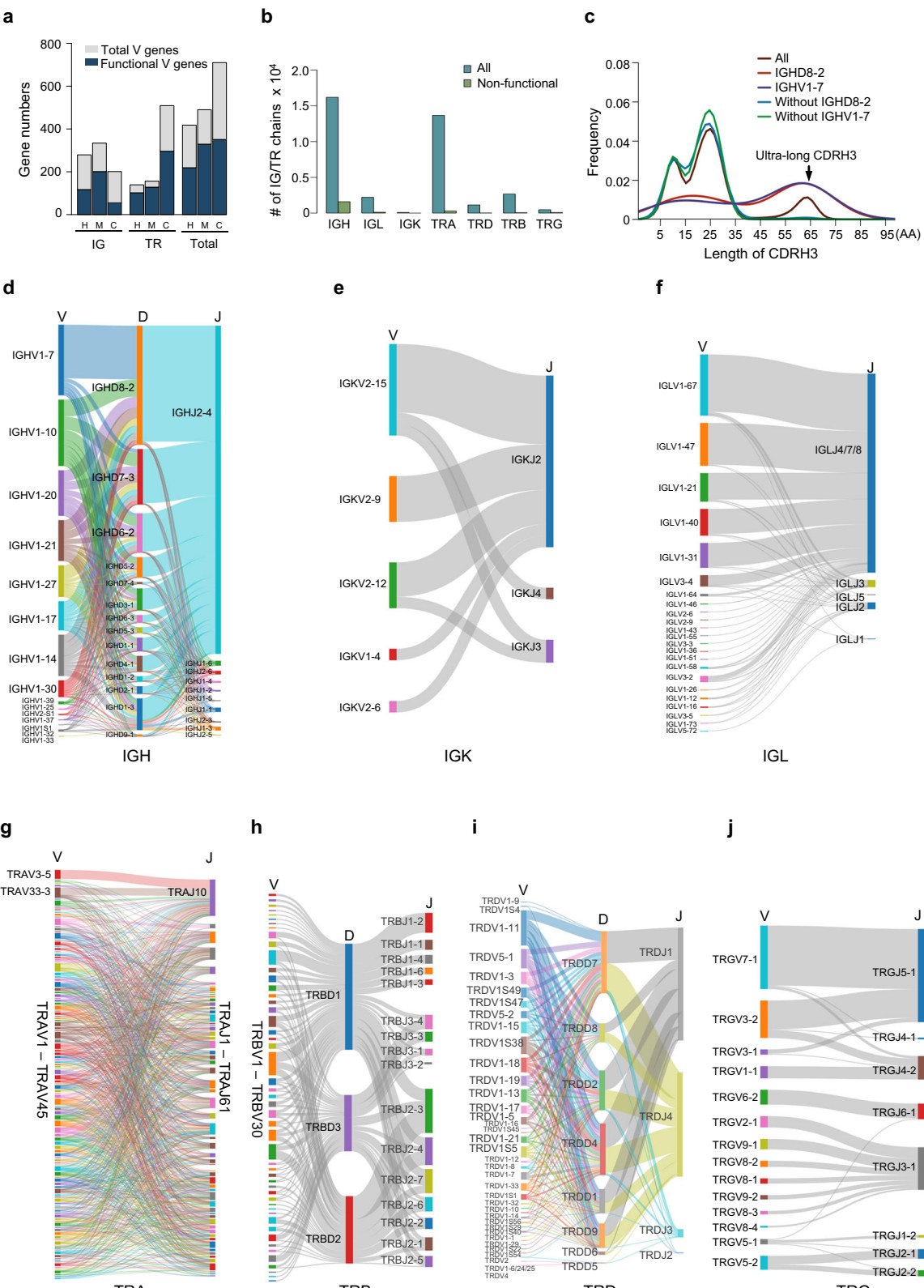

**Fig. 5 | Expression profiles of IG/TR genes. a** Numbers of V genes in human, mouse and cattle genome. H: human, M: mouse, C: cattle. **b** Numbers of full-length IG and TR sequences identified by PacBio HiFi sequencing. **c** Lengths of CDRH3 in IGH sequences. Sequence densities with or without IGHD8-2/IGHV1–7 were plotted individually and ultra-long CDRH3s were indicated. **d**–**j** Sankey diagram of V-(D)-J gene recombination preferences in IGs and TRs. IGs are composed of IGH (**d**), IGK (**e**) and IGL (**f**), and TRs are formed from TRA (**g**), TRB (**h**), TRD (**i**) and TRG (**j**).

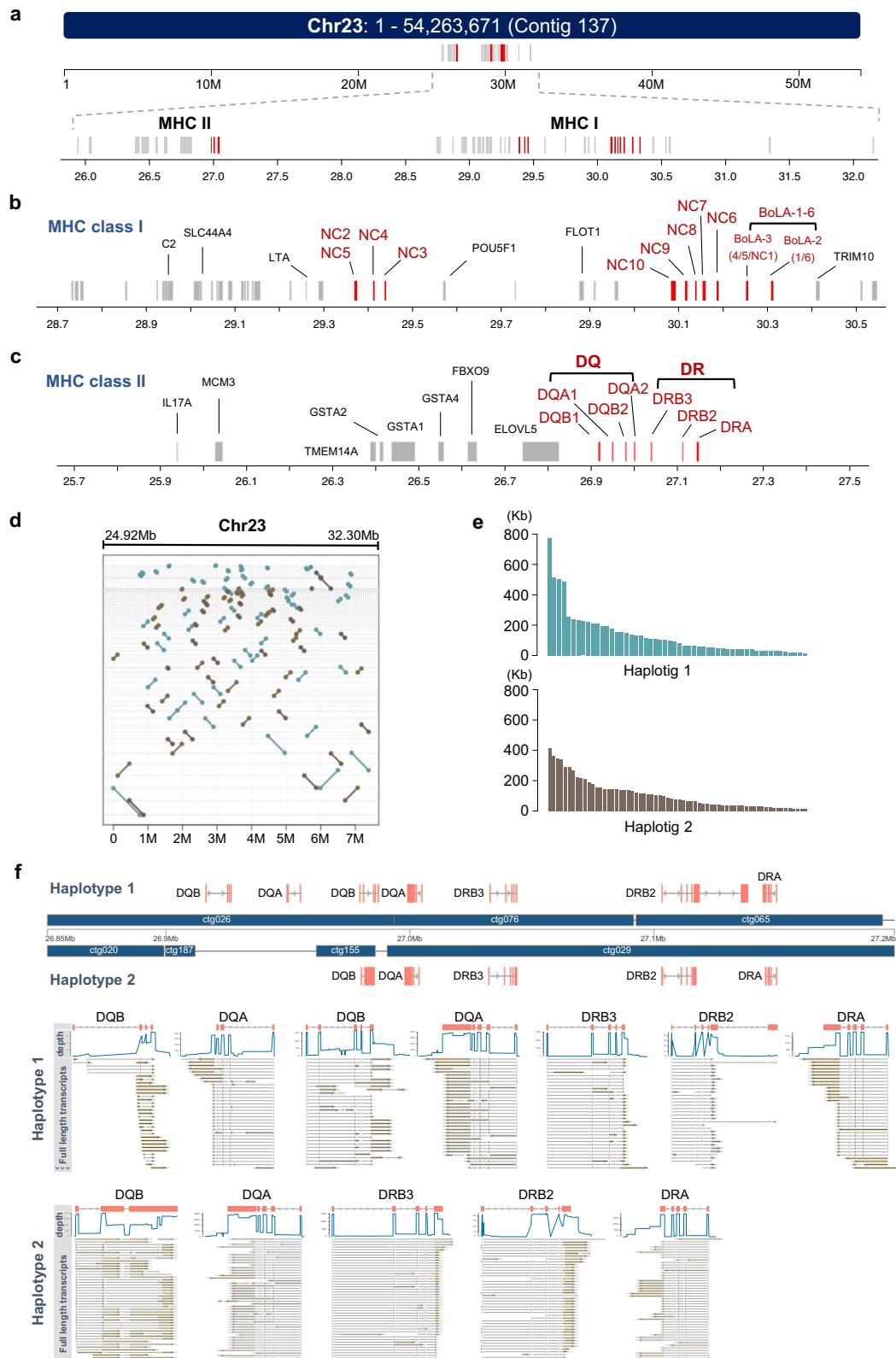

**Fig. 6 | MHC gene locus and haplotyping. a** Genomic coordinates of MHC locus in chromosome 23. Chromosome 23 consists of only one contig and MHC contains two separate genomic regions. **b, c** Detailed gene organizations of MHC class I (**b**) and class II (**c**). MHC genes were labeled in red. **d**–**f** Haplotyping of MHC genomic region. Genomic locations of two haplotigs within the MHC region (**d**). Length distributions of two haplotigs (**e**). Gene organization variation of two haplotigs and expression validation with full-length transcripts (**f**).

greater than that of human TR V genes. The reason why cattle genome contains more TR V genes than IG V genes remains obscure. As mucosal T cells play a central role in distinguishing dietary antigens and commensal bacteria from pathogens, the higher diversity of TR in cattle may be related with the gut-associated mucosal immunity[46]. Moreover, cattle have a large proportion of γδT cells that exhibit regulatory and antigen-presenting functions[47]. Further studies of cattle T cells shall shed light on the understanding of γδT cells, which remains elusive in humans due to their low abundance[48]. It is worth mentioning that the remarkably abundant and complex TR repertoire in cattle may serve as a natural resource pool for the screening of specific TRs with extraordinary therapeutic activity against human diseases, such as cancer.

In summary, the assembly, NCBA_BosT1.0, is a more complete and accurate reference of cattle genome, particularly the immune-genome, thereby facilitating further investigations of the immune system in cattle, and perhaps other mammals. Our data can be a blueprint for the final gapless telomere-to-telomere cattle genome assembly in the near future.

# Methods

### Study design

The goal of this study was to depict a complete immunogenomic landscape of cattle, including IGs, TRs and MHC. To achieve this goal, we assembled a cattle genome from de novo using ultra-long nanopore sequencing technology with other advanced technologies. All the genomic loci mentioned above were firstly gaplessly assembled and then gene structures and functionality were annotated following the IMGT criteria. We also profiled the immune repertoires with full-length PacBio HiFi sequencing. Detailed descriptions of the experimental procedures and specific analysis are provided below. Sample sizes, number of independent replicates, and statistical tests are indicated in the figure legends.

### Sample collection and genomic DNA isolation

The Holstein cattle of high genetic merit were mated to produce an elite fetus that was recovered at day 60. The cattle fibroblast cells were isolated from this fetus via disaggregation of all tissue excluding the viscera and limbs, and cultured in Dulbecco's Modified Eagle's Medium (DMEM; Gibco, Grand Island, New York, USA) supplemented with 10% fetal bovine serum (FBS; Gibco, Grand Island, New York, USA) at 37.5 °C in an atmosphere of 5% $CO_2$ and humidified air. The genomic DNA was extracted using the QIAGEN Genomic-tips kit (Cat. No.10223, QIAGEN, Valencia, CA, USA) according to the manufacturer's instructions.

### Ultra-long library construction and sequencing

To obtain ultra-long reads, only the large DNA fragments were selectively recovered with BluePippin, followed with end repair and dA-tailing (NEBNext Module, MA, USA). After careful purification, the adapter ligation was performed with SQK-LSK109 ligation kit (Oxford Nanopore Technologies, Oxford, UK). The final product was quantified by fluorometry (Qubit) to ensure >500 ng DNA was retained and sequenced on the Oxford Nanopore PromethION platform. ONT ultra-long reads were generated by Grandomics Biosciences company and only reads with a minimum mean quality score of 7 were kept for the following assembly.

### Hi-C library construction and sequencing

The Hi-C experiment was performed exactly following the in situ Hi-C method[49]. Briefly, the cross-linked cells were lysed and digested with MboI, filled with biotin-dATP, ligated with T4 DNA ligase and reverse crosslinked. Then the biotin-labeled DNA was enriched and sequenced with Illumina sequencing platforms following the manufacturer's instructions. The read mapping, quality control and matrix building were performed with HiC-Pro[50].

### De novo genome assembly

The de novo assembly of ONT ultra-long reads was performed with NextDenovo (https://github.com/Nextomics/NextDenovo). The reads were first self-corrected to generate consensus sequences with Next-Correct module and then assembled into preliminary assembly with NextGraph module. To correct the preliminary assembly, the original ONT reads and PacBio CCS reads were mapped with minimap2[51] and corrected with Racon[52] with default parameters for three iterations. Then the Illumina reads were used to polish the corrected assembly with NextPolish[53] for 4 iterations to generate the final polished genome assembly. The polished assembly was used as reference for the de novo assembly of BioNano data to generate scaffolds. For Hi-C data, LACHESIS[54] was used to cluster, order and direct the scaffolds to generate the final chromosomes of the assembly.

### Assembly evaluation

BUSCO 3.1.0 (-l mammalia_odb9 -g genome)[55] was used to evaluate the genome completeness based on included gene numbers, and CEGMA v2 was used to assess the assembly based on included eukaryotic protein core families with default parameters. Sequence accuracy was assessed based on the total number of homozygous SNPs identified by Illumina reads mapped to the assembly. Exogenous pollution was assessed based on the distributions of GC-depth and reads coverage.

### Genome annotation

For repeated sequences, TRF[56] was used to identify tandem repeats and RepeatMasker[57] was used to identify transposon-based elements. For gene structures, PASA was used to predict gene coordinates based on Illumina RNA-Seq data, GeMoMa was used to predict gene coordinates based on protein sequences of proximal species, and GeneMark-ST was used to predict genes from de novo. The three gene sets were integrated into an initial gene set with EVM[58] and finally to a clean gene set by removing genes containing transposable elements with Transposon-PSI (http://transposonpsi.sourceforge.net/). For further gene function annotation, the protein sequences of the predicted gene set were searched against several databases to predict their functions, including Non-Redundant Protein Database (NR, https://ftp.ncbi.nlm.nih.gov/blast/db/), Kyoto Encyclopedia of Gene and Genome (KEGG, https://www.genome.jp/kegg/), euKaryotic Orthologous Groups of protein (KOG, https://www.ncbi.nlm.nih.gov/research/cog), Inter-ProScan GO database (https://github.com/ebi-pf-team/interproscan), and Swiss-Prot database (https://www.expasy.org/resources/uniprotkb-swiss-prot).

### Genome comparison and gap filling

Genome sequence comparison between ARS-UCD1.2 and the new assembly was performed with LASTZ at chromosomal level. To fill the gaps of ARS-UCD1.2, 10 kb sequences up and downstream of the gap sites in chromosomes were fetched and aligned to the new assembly with minimap2[51]. Only alignments with >90% identity were kept and the alignment results of pair of gap sequences were manually checked to ensure the gap loci were within one contig. The unplaced scaffolds were first split up at gap loci and then aligned to the new assembly with minimap2. The scaffolds were reported if >50% alignment identity and located within one contig.

### Telomeres and satDNA annotation

To get the loci and lengths of telomeres, all short tandem repeats were identified with TRF[56] within the new assembly. Then short tandem repeats of TTAGGG were identified and only these located at the end of chromosomes were kept as telomeres. To annotate satDNAs, 57 nucleotide sequences belong to 8 satDNA classes were collected from NCBI and a blast database was built containing these 57 sequences. The sequence similarities among these 57 sequences were analyzed with blast. Then all short tandem repeats were identified with TRF, and their

pattern sequences were used to construct a fasta file and then blasted against the satDNA database. The alignments were filtered with >80% sequence identity and the locations and copy numbers were merged from TRF results.

### IG and TCR gene annotation

All cattle IG and TCR gene sequences were downloaded from IMGT database[59]. The gene sequences were aligned to the new assembly with bowtie2[60], and the alignment results were merged and manually checked according to their subgroups (IGH, IGK, IGL, TRA, TRB, TRD and TRG) to make sure the gene clusters were seamlessly assembled. For each locus, all candidate variable (V), diversity (D), joining (J), and constant (C) genes were manually annotated according to the following criteria (Supplementary Fig. 8). Manual annotations were validated by four independent people.

### Phylogenetic analysis of V genes

Functional IG and TR V gene sequences of human and mouse were downloaded from IMGT database[59]. Functional cattle V genes were retrieved from NCBA_BosT1.0. Multiple sequence alignment was performed with Clustal Omega[61], and the outputs were visualized using the Interactive Tree of Life software[62].

### MHC gene annotation and haplotyping

Totally 713 BoLA gene allele sequences were downloaded from IPD-MHC database[63] (https://www.ebi.ac.uk/ipd/mhc/). The BoLA gene sequences were aligned to the new assembly with bowtie2[60], and the genomic location and order of MHC Class I/II genes were manually confirmed with alignments > MAPQ 20. For haplotyping, PacBio HiFi reads that mapped to MHC region were retrieved and used for variant calling, followed with SNP genotyping with WhatShap[64]. Then the genotyped reads were separately retrieved and assembled into haplotypes with Canu PacBio-HiFi mode[65].

### MHC gene transcription validation

Gene sequences of MHC II were aligned to the genome assembly and MHC haplotypes to give all possible MHC II gene loci (step 1). Then, PacBio full-length transcripts were aligned to MHC II genes to give all possible transcripts for each gene (step 2). Next, potential MHC II gene transcripts obtained from step 2 were aligned to MHC haplotypes, and a transcript was treated as credible and kept only if the transcript was fully aligned to any gene locus obtained in step 1 with a minimum overall sequence identity of 95% (step 3). Finally, all potential MHC gene loci (obtained in step 1) were checked manually, and an MHC gene locus was treated as veritable if covered by multiple full-length transcript (obtained in step3) and the gene structure was determined by integrating full-length transcripts alignment information (step 4).

### Full-length IG/TR transcriptome profiling

Cell isolation, RNA extraction and library preparation were performed independently on four adult Holstein cows. For each cattle, 10 ml whole blood were collected, followed by PBMC cells isolation and total RNA extraction. cDNAs were generated and amplified using the Clontech SMARTer cDNA synthesis kit. The amplified cDNAs were firstly selected by BluePippin System, and then subjected to SMRTbell library construction with unique barcode. The libraries were then pulled together and sequenced with PacBio Biosciences RS II sequencer. The sequencing data were merged and analyzed by MiXCR[66] with default parameters and the Sankey diagram were performed with Sankey-MATIC software (https://sankeymatic.com).

### Statistical analysis

Quantification methods and statistical analysis for each of the separate and integrated analyses are described and referenced in their respective Method subsections in detail.

### Reporting summary

Further information on research design is available in the Nature Portfolio Reporting Summary linked to this article.

## Data availability

The raw sequence data reported in this paper have been deposited in the Genome Sequence Archive in National Genomics Data Center[67], China National Center for Bioinformation under accession number CRA006888 that can be publicly accessible at https://ngdc.cncb.ac.cn/gsa. The whole genome assembly and annotation data of NCBA_BosT1.0 have been deposited in the Genome Warehouse in National Genomics Data Center under accession number GWHBISA00000000 (https://ngdc.cncb.ac.cn/gwh/Assembly/25200/show) that can be publicly accessible at https://ngdc.cncb.ac.cn/gwh. All other data are available in the article and its Supplementary files or from the corresponding author upon request. Source data are provided with this paper.

## Code availability

The related codes developed for data analysis and visualization are available at GitHub (https://github.com/TintingLi/cattleGenome) and Zenodo (https://doi.org/10.5281/zenodo.8334734).

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

## Acknowledgements

This work was supported by grants from the National Key Research and Development Program of China (No. 2020YFA0707702 to T.L. and No. 2020YFA0707703 to T.Z.) and China National Natural Science Foundation (No. 81925017 to T.L. and No. 81872153 to T.Z.).

## Author contributions

T.L., T.-T.L., and T.Z. conceptualized and designed the project. T.X., Z.-L.S., M.W., and F.-R.D. prepared the cells. T.-T.L., J.-Q.W., and T.X. analyzed the data. T.-T.L., J.-Q.W., T.X., H.H., S.J., J.L., J.W., G.Y., J.-N.F., Y.-P.D., and J.P. annotated the IG, TR, and MHC genes. T.L., T.-T.L., T.X., and X.-M.Z. wrote the manuscript with the help of all authors.

## Competing interests

All the authors declare no competing interests.

## Ethics approval

The procedures were performed in strict accordance with the Guide for the Care and Use of Laboratory Animals. All the animal work in this study was approved by the ethics committee of China Agricultural University (No. 2017-04-11).
