## [Peer Review File · Nature Communications]

De novo Genome Assembly Depicts the Immune Genomic Characteristics of CattleREVIEWER COMMENTS

Reviewer #1 (Remarks to the Author):

The paper describes a multi-platform approach to achieving a more complete characterisation of the bovine genome. The authors use the more comprehensive detailing of the immunogenetics loci as a key example of how the highly contiguous genome enhances our understanding of bovine biology.

The report represents a substantial body of work that undoubtedly has value, however there are a number of issues that need to be resolved prior to re-consideration of the possible acceptance for publication. I should emphasize that my experience relates more to the immunogenetics than the genome assembly portion of the work.

1 - English language - this is the most minor of the points I am raising. In several places through the manuscript the language is 'awkward' which distracts from the enjoyment of the reading but is generally not leading to mis-representation of the information presented. However, I do feel sorry for the 'irrelevant people' on line 448! It would be good, if re-submission is invited, to ensure that there is a robust editorial review of the text.

2 - Scientific queries on the immunogenetics loci. There are a number of pieces of data that are interesting for not adhering to paradigms that the authors don't comment on as being notable. Examples include Figure 5h - the paradigm would be that TRB chains that utilise the DJ2 gene segment should recombine with J genes of the J2 cluster only - however in this figure there appears to be recombination of DJ2 with both J1 and J3 cluster J genes as well - could the authors comment on this? Similarly in Figure 6f the authors' data suggests the presence of 2 x DRB3 loci, 2 x DQB loci but only 1 x DQA locus. This is an intriguing observation as it is generally held that although there may be multiple DRB loci (as implied by there being a DRB3) there should only be a single DRB3 locus. Also - it is generally inferred that the DQA and DQB loci, when duplicated are usually duplicated together - so in general you would expect either [DQA/DQB]_{x1} or [DQA/DQB]_{x2} but not [DQA_{x1}, DQB_{x2}]. Of course this MHC haplotype in the animal selected may genuinely have this genotypic structure (the amount of analysed data in this research area is limited) but it is of some concern that these unusual features didn't attract comment from the authors.

3 - Literature references - in general the use of literature was not appropriate. A number of groups have published data on bovine MHC, Ig and TCR repertoires and diversity in cattle -

virtually none of it was cited. For example there have been multiple papers citing the large expansion of bovine TCR repertoires (and so the observations about this in the paper aren't really that novel; it should also be noted that from an immunogenetics perspective the data presented in Figure 5 don't really describe the Ig and TCR repertoires - only the V(D)J permutations which is largely a predictable product of the number of V(D)J genes available for recombination - the text in the manuscript is inconsistent in how it refers to this). A better appreciation of the literature would have helped construct a better narrative. Similarly to claim characterisation of the complete immunogenomic repertoire, the authors may also need to consider inclusion of other highly polymorphic loci that have been difficult to authoritatively characterise genomically, such as the NK receptor loci (LRC and NK gene complex), in the analysis. For my perspective the issues here and in 2 above are those that are most critical to address

4 - Details in methodology - in some areas the description of the methodology is insufficient. For example the description of the 'Full length immune repertoires' had some components that need further explanation - why was cDNA derived from 4 animals? Was this cDNA pooled prior to sequencing or after (and so could subsequently be de-multiplexed)? Don't quite understand what has happened and why. Think it should be possible to improve on the descriptions with minor effort.

Reviewer #2 (Remarks to the Author):

Summary:

This manuscript reports an extraordinarily large genomics study of cattle. The authors overcame the difficulties encountered by others in earlier studies seeking to obtain full genome sequence. Short read sequences simply could not provide the needed data. The authors were successful in obtaining high quality long reads using ultra-long-read nanopore sequencing methods. The accuracy of the data were bolstered with data from earlier sequencing methods including PacBio HiFi and Illumina sequencing to verify regions of sequence. The result is an essentially a new, fully assembled sequence for the genome of cattle. The sequence in this dataset is far more cohesive than the previously published assembly. The work represents a major advance with far fewer breaks remaining in this newly assembled sequence. It is a significant achievement. Rather than attempt to report

the full genome contents in detail, the authors chose to focus on three parts of the cattle immune system. They were able to seamlessly assemble full sequences for the genomic regions in which the cattle B-cell receptors (immunoglobulin, IG), T-cell receptors (TR) and the major histocompatibility complex (MHC) molecules are encoded. These sequences are fully annotated and are discussed with respect to immune functions.

Writing a manuscript for such a huge quantity of information is challenging. The authors were successful in developing a way in which the findings are not weighted down by the huge content. The figures provide clear and useful illustrations. I was happy to find that the authors succeeded in producing a highly readable report.

Concerns:

1) The title (line 1): “The complete immunogenomic landscape of cattle” is not a particularly appealing title. It may be an overstatement in that there are surely other genes not in these three loci that contribute to immunity that are not in the details reported. The authors might want to work to find the words to describe the report more accurately.

“Immunogenomic landscape” also suggests one will come away with an “image” of this portion of the cattle genome. The data set is of such complexity that one does not come away with a “landscape image”. Rather one has multiple pictures of the complexity and intricacy of each gene region. The title is indeed important and difficult to get just right. Perhaps the title can be improved. It would be good to give readers information in the title that identifies the significant advance reported.

2) There are places where the understandable enthusiasm of the authors shows through. This occurs in multiple places where the writing goes over the mark for formal scientific writing. To say that the sequence “exquisitely annotated” (line 28) is a little off. Maybe “precisely annotated” or something like that would be better. The use of “excitingly” (line 52) reflects the authors enthusiasm. Maybe it is not necessary.

3) The quality of the writing is generally quite good. There is need some minor editing. For examples, “promoted a prosperity of genome assemblies” (line 64) misses the mark. Maybe it should be “an abundance of assemblies”. The “telomere-to-telomere assembly of specific chromosome” (line 66) seems off. Do the authors mean full assemblies of individual chromosomes from telomere to telomere? The use of “privileges” (line 74) is not quite right. Do the authors mean there are advantages in using cattle to study human infectious diseases? Individuals have “achieved great success in the complete assembly....” (line 301)

not "sequencing-technology progress". These are examples of unconventional word usage. Just a little editing is needed.

Reviewer #3 (Remarks to the Author):

This manuscript by Ting-Ting et al is an important advance in our understanding of the structure of the cattle genome, with a focus on immunogenetic regions that are notoriously difficult to assemble, even with longer reads due to their repetitive nature and polymorphism. The genomic structure can have important effects of expression and rearrangement during an immune response, there this is important as the age of genome editing in livestock is upon us. I think this is the first time that the TRA/TRD locus has been contiguously assembled in cattle for example, a significant achievement.

However, I do have some major concerns overall. One suggestion would be to concentrate more on the assembly that the detail contained within the immunogenetic loci as this creates confusion and is often misleading.

1. This manuscript attempts to both describe the genome assembly and immunogenetic loci, but the former is not accomplished to any great extent. I agree that as a theme to discuss the contiguity of a genome this is a useful benchmark. However, it is clear that the authors are not experts. Almost all the contemporary literature is ignored for MHC, TCR and BCR, much of which is high quality and actually confirms the structure of the genome presented. The current ARS-UCD1.2 assembly is not even referenced, and complete haplotypes for many of the above regions are already published.

2. The importance of immunogenetic loci, particularly germline encoded, is in the polymorphism of each alleles, alongside the structure. For example, MHC haplotypes are defined by both in tandem. Therefore the level of detail of sequence and comparison across all the loci describes precludes publication if the focus is to remain on this level of immunogenetics.

3. Linked to the above, the sequengin error rate was mentioned for PacBio, but I didn't see

it discussed at all for MinION data this is the basis of the reference used. Considering the focus on allele polymorphism and relatively low coverage this has to be a major focus of the paper, quantified and evidenced.

More minor comments

4. Line 158 and Figure 2; to clarify the situation with IGH in ARS-UCD1.2, much of the IGH is incorrectly assembled to Chromosomes 20 (positions: 71881100-71974595) and 21 (positions: 1-411439) and some of it is unplaced (e.g., NKLS02001456.1). The text and figure should reflect this as it is currently misleading.

5. Lines 177-8; While novel loci could be given provisional names in the manuscript, renaming gene segments that already have established IMGT names will create confusion. It would be more useful to retain the established IMGT nomenclature than to re-invent it. IMGT nomenclature is pseudo-positional – it is not strictly required to maintain positional numbering in order to account for gene content variation. For example: PMID: 24934119

6. Lines 182-3 (example); there are several places in the text that suggest that the newly described assembly “corrects” the reference assembly. Care should be taken when making these statements since the two genomes are derived from different individuals. It is quite possible (and perhaps expected) that haplotypic variation in gene content exists in these repetitive immune-related gene complexes. Similarly, the new assembly is not simply a new “version” (line 192), since it is not an update to the existing reference assembly (i.e., it is an entirely different individual!)

7. Line 273; Ref 32 describes the felid MHC. A more appropriate reference for the cattle MHC, and which established the NC6-NC10 nomenclature, is: PMID: 34802191

8. Line 270; Ref 8 is for the human MHC. A more appropriate reference for cattle (MHC I) is: PMID: 24934119. Also, the lack of DP genes in cattle was already reported long ago (PMID: 2891610)

9. Line 448; “... validated by four irrelevant people.” I think “independent” is maybe

intended here instead of “irrelevant”?

10. Line 456; please include a citation for IPD-MHC (PMID: 2891610)

Point-by-Point Response:

Reviewer #1:

The paper describes a multi-platform approach to achieving a more complete characterization of the bovine genome. The authors use the more comprehensive detailing of the immunogenetics loci as a key example of how the highly contiguous genome enhances our understanding of bovine biology.

The report represents a substantial body of work that undoubtedly has value, however there are a number of issues that need to be resolved prior to re-consideration of the possible acceptance for publication. I should emphasize that my experience relates more to the immunogenetics than the genome assembly portion of the work.

Response: We greatly appreciate the reviewer's encouraging comments and constructive suggestions on our manuscript. Following these suggestions, we carried out additional comprehensive analysis of our data and revised our manuscript accordingly.

Concerns:

1 - English language - this is the most minor of the points I am raising. In several places through the manuscript the language is 'awkward' which distracts from the enjoyment of the reading but is generally not leading to mis-representation of the information presented. However, I do feel sorry for the 'irrelevant people' on line 448! It would be good, if re-submission is invited, to ensure that there is a robust editorial review of the text.

Response: We apologize for the language polishing issues and grammar errors present in our manuscript. As the reviewer mentioned, we replaced "irrelevant" with "independent". Besides, we made a substantial editing of the text to improve the clarity and readability.

2 - Scientific queries on the immunogenetics loci. There are a number of pieces of data that are interesting for not adhering to paradigms that the authors don't comment on as being notable. Examples include Figure 5h - the paradigm would be that TRB chains that utilize the DJ2 gene segment should recombine with J genes of the J2 cluster only

- however in this figure there appears to be recombination of DJ2 with both J1 and J3 cluster J genes as well - could the authors comment on this? Similarly in Figure 6f the authors' data suggests the presence of 2 x DRB3 loci, 2 x DQB loci but only 1 x DQA locus. This is an intriguing observation as it is generally held that although there may be multiple DRB loci (as implied by there being a DRB3) there should only be a single DRB3 locus. Also - it is generally inferred that the DQA and DQB loci, when duplicated are usually duplicated together - so in general you would expect either [DQA/DQB]x1 or [DQA/DQB]x2 but not [DQAx1, DQBx2]. Of course this MHC haplotype in the animal selected may genuinely have this genotypic structure (the amount of analysed data in this research area is limited) but it is of some concern that these unusual features didn't attract comment from the authors.

Response: We thank the reviewer for the important suggestions. Following these suggestions, we carried out additional analysis and revised our manuscript accordingly.

(1) In TRB locus, the V gene segments are followed by [D-J-C]₁-[D-J-C]₃-[D-J-C]₂ tandem gene segment clusters (**Figure. 4b**). The reviewer pointed out that DJ2 gene segment should recombine with J genes of the J2 cluster only, while there appears to be recombination of DJ2 with both J1 and J3 genes in our original manuscript (**Figure. 5h**).

After detailed retrospect of the immune repertoire data analysis process, we found that the problem raised from the high sequence similarity of the three TRBD gene segments (**Figure. R1A**). Thus, the inference of D segment from a TRB transcript cannot always be precisely achieved (an example is shown in **Figure. R1B**). In our original analysis, the D segment with the highest alignment score was selected if multiple D segments were aligned. Following the reviewer's suggestion, we analyzed the alignment results of TRBD segments, and found that only 56.1% TRB transcripts were deduced with one unique TRBD segment, 25.5% TRB transcripts can be aligned to multiple TRBD segments, and 18.4% TRB transcripts cannot be aligned to TRBD segments at all, which can be attributed to the V(D)J rearrangement and somatic mutation process (**Figure. R1C**). D segment inference from IGH and TRD transcripts showed the similar results (**Figure. R1C**). In contrary, the J segment showed very high sequence specificity (**Figure. R1D**). Next, we re-analyzed the D-J pairs in TRBD transcripts that uniquely aligned to one TRBD gene (**Figure. R1C**), and found that TRBD-TRBJ segments can only be paired in sequential order. Moreover, we noticed that TRBV30 is located downstream of DJC

Figure. R1 Systematic analysis of D-J recombination of TRB transcripts. (A) Multiple sequence alignment of three TRBD gene segments. (B) An example of PacBio full-length TRB transcripts, with TRBV, D and J gene segments labeled with different colors. Determining the TRBD gene is challenging based on the sequence alignment. (C-D) Statistics of D gene segment usages (C) and J gene segment usages (D) from full-length transcripts. (E) The distribution of unique pairwise combinations of D-J gene segments of TRB transcripts. (F) An example of PacBio full length TRB transcripts which was rearranged with TRBV30-TRBJ2-TRBC2 gene segments.

clusters in reverse transcriptional orientation (**Figure. 4b**), and the functionality of TRBV30 was further validated with full-length transcripts (**Figure. R1F**).

Besides TRB locus, there exist similar tandem gene clusters in other immune genomic loci that share the same issue, such as IGL and TRG. IGL consists of a [IGLJC]₁-[IGLJC]₂-[IGLJC]₃-[IGLJC]₄-[IGLJC]₅-[IGLJC]₆ gene cluster (**Figure. 2c and Supplementary Figure. 11a in our revised manuscript**). TRG1 consists of [TRGVJC]₅-[TRGVJC]₇-[TRGVJC]₃-[TRGVJC]₄ gene cluster while TRG2 consists of [TRGVJC]₁- [TRGVJC]₂-[TRGVJC]₆ gene cluster (**Figure. 4c and Supplementary Figure. 13c in our revised manuscript**). Given that the two TRG loci are over 30Mb distance away from each other, the V-J recombination between TRG1 and TRG2 is unlikely occurred. Based on above analysis, we corrected and updated the immune repertoire results of IGL (**Figure. 5f in our revised manuscript**), TRB (**Figure. 5h in our revised manuscript**) and TRG (**Figure. 5j in our revised manuscript**). IGK results were also updated according to the functionality annotation of IGKJ genes from IMGT database (**Figure. 5e in our revised manuscript**).

(2) Regarding the MHC locus, the reviewer pointed out that there should only be a single DRB3, but MHC haplotype 1 consists of two tandem DRB3 loci in our original manuscript (**Figure. 6f**). To fully address this point, we set up a pipeline integrated with MHC gene sequence mapping and gene full-length transcripts validation (**Figure. R2A**). Briefly, gene sequences of MHC II were aligned to the genome assembly and MHC haplotypes to give all possible MHC II gene loci (step 1). Then, PacBio full-length transcripts were aligned to MHC II genes to give all possible transcripts for each gene (step 2). Next, potential MHC II gene transcripts obtained from step 2 were aligned to MHC haplotypes, and a transcript was treated as credible and kept only if the transcript was fully aligned to any gene locus obtained in step 1 with a minimum overall sequence identity of 95% (step 3). Finally, all potential MHC gene loci (obtained in step 1) were checked manually, and an MHC gene locus was treated as veritable if covered by multiple full-length transcript (obtained in step3) and the gene structure was determined by integrating full-length transcripts alignment information (step 4).

Following this pipeline (**Figure. R2A**), the first DRB3 locus was validated by a large number of full-length transcripts, while the second DRB3 locus was determined as invalidated as only part of the DRB3 gene sequence were

Figure. R2 MHC gene loci validation and full-length transcripts alignment. (A) Analysis pipeline for validating MHC gene loci and structure in two haplotypes. (B) DRB3 gene locus validation with full-length transcripts following the criteria in (A). From top to bottom, the three rows represent graphic track of DRB3 gene model, read coverage of full length DRB3 transcripts and the detailed read alignments. (C) The complete MHC class II gene loci and structures in both haplotypes. The assembled contigs of haplotype are labeled in deep blue color. The validation of each gene with PacBio full-length transcripts mapping was demonstrated at the bottom.

aligned to this locus without full-length transcripts supporting its transcript activity (**Figure. R2B**). Additionally, we noticed that there exist multiple full-length transcripts that can only be partially aligned to DRB3 locus, but can be fully and accurately aligned to another position located between DRB3 and DRA, indicating the existence of an additional gene that shares sequence similarity to DRB3. After carefully comparing with the gene annotation information of cattle Refseq genome (<https://www.ncbi.nlm.nih.gov/genome/82>), we confirmed that it was a DRB2 locus (<https://www.ncbi.nlm.nih.gov/gene/538700>) between DRB3 and DRA (**Figure. R2C**), which was missed in the original manuscript (**Figure. 6f in our original manuscript**).

(3) The reviewer questioned our results regarding the duplication of the DQA and DQB loci in MHC haplotype 1. Typically, these loci are duplicated together ([DQA/DQB] \times 1 or [DQA/DQB] \times 2). In our original manuscript, we characterized two DQB loci and one DQA locus in MHC haplotype 1 (**Figure. 6f in our original manuscript**). Following the same pipeline described above, we further analyzed and verified all MHC class II gene loci and structures in both haplotypes. We ultimately confirmed the existence of two [DQA/DQB] loci in MHC haplotype 1. Our data suggested that there was one [DQA/DQB] locus in MHC haplotype 2 (**Figure. R2C**). The second DQA locus in MHC haplotype 1 was missed in our original manuscript as the alignment to DQA gene sequence was less than 90% and the PacBio full-length transcripts were not taken into consideration. We thank the reviewer for the important and valuable suggestions and instructions. Accordingly, we revised the main text (line 307-309, and line 497-507), and updated **Figure. 6c, 6f and Supplementary Figure. S15a** in our revised manuscript.

3 - Literature references - in general the use of literature was not appropriate. A number of groups have published data on bovine MHC, Ig and TCR repertoires and diversity in cattle - virtually none of it was cited. For example there have been multiple papers citing the large expansion of bovine TCR repertoires (and so the observations about this in the paper aren't really that novel; it should also be noted that from an immunogenetics perspective the data presented in Figure 5 don't really describe the Ig and TCR repertoires - only the V(D)J permutations which is largely a predictable product of the number of V(D)J genes available for recombination - the text in the manuscript is inconsistent in how it refers to this). A better appreciation of the literature

would have helped construct a better narrative. Similarly to claim characterization of the complete immunogenomic repertoire, the authors may also need to consider inclusion of other highly polymorphic loci that have been difficult to authoritatively characterise genomically, such as the NK receptor loci (LRC and NK gene complex), in the analysis. For my perspective the issues here and in 2 above are those that are most critical to address

Response: We appreciate the reviewer's valuable comments and suggestions.

(1) We apologize for the inadequate literature referencing of bovine immunogenetics. Following the reviewer's suggestions, we carefully reviewed the recent publications on MHC diversity and IG/TR repertoires in cattle, and incorporated the following important references in our revised manuscript:

MHC I and II diversities of cattle populations have been analyzed with high throughput sequencing methods (PMID:36680506, Silwamba, Vasoya et al. 2023; PMID: 34102036, Vasoya, Oliveira et al. 2021), as well as genetic diversity studies of specific genes, such as BoLA-DRB3 (PMID: 33094557, Giovambattista, Takeshima et al. 2020; PMID: 32867670, Giovambattista, Moe et al. 2020). For database IPD-MHC, we added reference (PMID: 27899604, Maccari, Robinson et al. 2017). We also included publications regarding bovine IG repertoires (PMID: 34464839, Oyola, Henson et al. 2021), and cattle light chain usage (PMID: 8968166, Arun, Breuer et al. 1996). For bovine TR repertoires, PBMC $\gamma\delta$ TCR repertoires in cattle have been studied using high-throughput sequencing approaches (PMID: 36426936, Gillespie, Loonie et al. 2022), and our results are consistent with this study (**Figure. 5i and 5j in the revised manuscript**). The TRA/D locus contains over 400 V genes and encodes V genes without CDR2 (PMID: 19568741, Reinink and Van Rhijn 2009) and TRA/D genes are comparative analyzed by IMGT (PMID: 33379283, Pegorier, Bertignac et al. 2020).

As reviewer 3 also suggested, we added references related to the genomic structures of MHC/IG/TR (**MHC haplotypes:** PMID: 26227296, Schwartz and Hammond 2015; PMID: 34802191, Schwartz, Maccari et al. 2022; PMID: 33685702, Bakshy, Heimeier et al. 2021; **ARS-UCD1.2 assembly:** PMID: 32191811, Rosen, Bickhart et al. 2020; **IG/TR structures:** PMID: 22497300, Stein, Diesterbeck et al. 2012; PMID: 19393068, Connelley, Aerts et al. 2009; and PMID: 17141331, Conrad, Mawer et al. 2007).

With the adding of above references, we revised the main text (line 75-77; line 83-85; line 173, line 211, line 263, line 277, line 283-287 and line 295). That would be much appreciated if the reviewer could kindly point out any areas where we may have fallen short in literature referencing.

(2) The reviewer pointed out that “*the data presented in Figure 5 don’t really describe the Ig and TCR repertoires - only the V(D)J permutations which is largely a predictable product of the number of V(D)J genes available for recombination*”. We apologize for the unclear description about ‘Full length immune repertoires’ in our original manuscript. We isolated PBMCs from cattle and performed PacBio HiFi full-length transcriptome sequencing. With these data, we profiled the diversity of the full-length IG/TR transcripts. By doing this, we further verified the functionality of different V/D/J genes that annotated in our assembly. Following the reviewer’s suggestion, we changed the subtitle of the corresponding part of the result from “Full-length immune repertoires profiling” to “Full-length IG/TR transcriptome profiling”.

(3) According to the reviewer's suggestions, we depicted the genomic structures of the NK receptor loci (LRC and NK gene complex) which are of high polymorphic (**Figure. R3**). Both loci were seamlessly assembled in NCBA1.0. In terms of NK gene complex, the global gene structures exhibited a high degree of similarity between NCBA1.0 and ARS-UCD1.2, except a notable difference arises in the gene numbers of KLRC1 and its nearby homologs, as illustrated in **Figure R3A**. NCBA1.0 consists of one KLRC1 gene and five nearby highly similar genes (KLRC1-[2-6]) while ARS-UCD1.2 contains one KLRC1 gene and two KLRC1 similar genes (LOC100847738 and LOC100336869). For LRC gene complex, NCBA1.0 consists of 17 KIR genes in total, including one KIR2DL gene (2DL5A), two KIR2DS genes, eight KIR3DS genes and six KIR3DL genes (**Figure. R3B**), which was significantly more than ARS-UCD1.2. We included these data as **Supplementary Figure 16** in our revised manuscript and edited the text accordingly (line 315-329).

4 - Details in methodology - in some areas the description of the methodology is insufficient. For example, the description of the 'Full length immune repertoires' had some components that need further explanation - why was cDNA derived from 4 animals? Was this cDNA pooled prior to sequencing or after (and so could subsequently be de-multiplexed)? Don't quite understand what has happened and why. Think it should be possible to improve on the descriptions with minor effort.

Response: Regarding the 'Full length immune repertoires', we changed the subtitle of the corresponding part of the result from "Full-length immune repertoires profiling" to "Full-length IG/TR transcriptome profiling". We isolated PBMCs from four holstein cows and performed full-length RNA-seq experiments individually. The four samples from different animal serve as biological repeats. By performing PacBio HiFi full-length transcriptome sequencing, we profiled the diversity of the full-length IG/TR transcripts. By doing this, we further verified the functionality of different V/D/J genes that annotated in our assembly.

We apologize for our insufficient description in methodology. Following the reviewer's suggestions, we revised the Methods section and improved the clarity by adding experimental details (line 508-517 in our revised manuscript).

Reviewer #2:

This manuscript reports an extraordinarily large genomics study of cattle. The authors overcame the difficulties encountered by others in earlier studies seeking to obtain full genome sequence. Short read sequences simply could not provide the needed data. The authors were successful in obtaining high quality long reads using ultra-long-read nanopore sequencing methods. The accuracy of the data were bolstered with data from earlier sequencing methods including PacBio HiFi and Illumina sequencing to verify regions of sequence. The result is an essentially a new, fully assembled sequence for the genome of cattle. The sequence in this dataset is far more cohesive than the previously published assembly. The work represents a major advance with far fewer breaks remaining in this newly assembled sequence. It is a significant achievement. Rather than attempt to report the full genome contents in detail, the authors chose to focus on three parts of the cattle immune system. They were able to seamlessly assemble full sequences for the genomic regions in which the cattle B-cell receptors (immunoglobulin, IG), T-cell receptors (TR) and the major histocompatibility complex (MHC) molecules are encoded. These sequences are fully annotated and are discussed with respect to immune functions.

Writing a manuscript for such a huge quantity of information is challenging. The authors were successful in developing a way in which the findings are not weighted down by the huge content. The figures provide clear and useful illustrations. I was happy to find that the authors succeeded in producing a highly readable report.

Response: We greatly appreciate the reviewer's encouraging comments and important suggestions to improve our work. Following the reviewer's suggestions, we have carefully revised our manuscript, polished the language and addressed concerns with new data and discussion as detailed below.

Concerns:

1) The title (line 1): "The complete immunogenomic landscape of cattle" is not a particularly appealing title. It may be an overstatement in that there are surely other genes not in these three loci that contribute to immunity that are not in the details reported. The authors might want to work to find the words to described the report more accurately. "Immunogenomic landscape" also suggests one will come away with an "image" of this portion of the cattle genome. The data set is of such complexity that one does not come away with a "landscape image". Rather one has multiple pictures

of the complexity and intricacy of each gene region. The title is indeed important and difficult to get just right. Perhaps the title can be improved. It would be good to give readers information in the title that identifies the significant advance reported.

Response: We greatly appreciate the reviewer's suggestion. Following the reviewer's suggestion, we drafted a new title of our manuscript as "De novo Genome Assembly Depicts the Immune Genomic Terrain of Cattle". We would be grateful for the reviewer's further instruction.

2) There are places where the understandable enthusiasm of the authors shows through. This occurs in multiple places where the writing goes over the mark for formal scientific writing. To say that the sequence "exquisitely annotated" (line 28) is a little off. Maybe "precisely annotated" or something like that would be better. The use of "excitingly" (line 52) reflects the authors enthusiasm. Maybe it is not necessary.

Response: We are very grateful for the reviewer's valuable comments and suggestions regarding the scientific writing. We have carefully reviewed the entire manuscript and made appropriate revisions as follows:

In line 28 of the revised manuscript: we revised "exquisitely annotated" to "precisely annotated". And in line 53, we revised "Excitingly" to "Notably".

Moreover, in line 131 of the revised manuscript, instead of original statement "The NCBA1.0 assembly showed tremendous sequence integrity", we rephrased it to "The NCBA1.0 assembly demonstrated impressive sequence integrity".

3) The quality of the writing is generally quite good. There is need some minor editing. For examples, "promoted a prosperity of genome assemblies" (line 64) misses the mark. Maybe it should be "an abundance of assemblies". The "telomere-to-telomere assembly of specific chromosome" (line 66) seems off. Do the authors mean full assemblies of individual chromosomes from telomere to telomere? The user of "privileges" (line 74) is not quite right. Do the authors mean there are advantages in using cattle to study human infectious diseases? Individuals have "achieved great success in the complete assembly...." (line 301) not "sequencing-technology progress". These are examples of unconventional word usage. Just a little editing is needed.

Response: We appreciate the reviewer's valuable comments.

Following the reviewer's suggestions, we amended our manuscript by polishing the language as follows:

(1) "promoted a prosperity of genome assemblies" revised as "promoted a prosperity of an abundance of genome assemblies" (line 65 in the revised manuscript).

(2) "telomere-to-telomere assembly of specific chromosome" were reported including chromosome X (PMID: 32663838, Miga, Koren et al. 2020) and chromosome 8 (PMID: 33828295, Logsdon, Vollger et al. 2021).

(3) "privileges" revised as "advantages" (line 77 in the revised manuscript).

(4) "achieved great success in the complete assembly of human genome" revised as "greatly promoted the complete assembly of human genome" (line 337 in the revised manuscript).

Additionally, we made several other revisions throughout the manuscript according to the reviewer's suggestions.

Reviewer #3:

This manuscript by Ting-Ting et al is an important advance in our understanding of the structure of the cattle genome, with a focus on immunogenetic regions that are notoriously difficult to assemble, even with longer reads due to their repetitive nature and polymorphism. The genomic structure can have important effects of expression and rearrangement during an immune response, there this is important as the age of genome editing in livestock is upon us. I think this is the first time that the TRA/TRD locus has been contiguously assembled in cattle for example, a significant achievement.

However, I do have some major concerns overall. One suggestion would be to concentrate more on the assembly than the detail contained within the immunogenetic loci as this creates confusion and is often misleading.

Response: The reviewer indicated that our study is both novel and significant. We appreciate the reviewer's encouraging comments and important suggestions to improve our work. Following their suggestions, we have carefully revised our manuscript, polished the language and addressed concerns with new data and discussion as detailed below, point by point.

Major comments :

1. This manuscript attempts to both describe the genome assembly and immunogenetic loci, but the former is not accomplished to any great extent. I agree that as a theme to discuss the contiguity of a genome this is a useful benchmark. However, it is clear that the authors are not experts. Almost all the contemporary literature is ignored for MHC, TCR and BCR, much of which is high quality and actually confirms the structure of the genome presented. The current ARS-UCD1.2 assembly is not even referenced, and complete haplotypes for many of the above regions are already published.

Response: We appreciate the reviewer's important comments and suggestions. We apologize for the inadequate literature referencing of bovine genome and immunogenetics. Following the reviewer's suggestions, we carefully reviewed the recent publications on bovine genome assembly, and MHC/IG/TR structures in cattle, and added the following important references in our revised manuscript:

The genomic structure of bovine MHC-Ia region was first sequenced and assembled with BAC libraries ([PMID: 26227296](https://pubmed.ncbi.nlm.nih.gov/26227296/), Schwartz and Hammond 2015), and the

gaps within the BAC-derived MHC assembly were closed in the current reference assembly, ARS-UCD1.2 (PMID: 32191811, Rosen, Bickhart et al. 2020). Accurate functional analyses based on the previously studies promote the establishment of MHC-I nomenclature, especially the NC6-NC10 genes (PMID: 34802191, Schwartz, Maccari et al. 2022). Assemble of full-length haplotypes of MHC also facilitates the development of polymorphic markers within the loci (PMID: 33685702, Bakshy, Heimeier et al. 2021).

For bovine B-cell receptors, the IGH genomic locus was assembled and described based on BAC library sequencing (PMID: 27053761, Ma, Qin et al. 2016), and the J and C gene regions of IGK were compared in different cattle breeds (PMID: 22497300, Stein, Diesterbeck et al. 2012). For bovine T-cell receptors, the TRA/TAD locus contains over 400 V genes and encodes V genes without CDR2 (PMID: 19568741, Reinink and Van Rhijn 2009) and is comparative analyzed by IMGT (PMID: 33379283, Pegorier, Bertignac et al. 2020), genomic analysis reveals extensive gene duplication with in the bovine TRB locus (PMID: 19393068, Connelley, Aerts et al. 2009) and the genomic sequence of TRG loci localized the TRGC5 cassette (PMID: 17141331, Conrad, Mawer et al. 2007).

The added references:

We updated the main text between line 75-77 in our revised manuscript, and added the references related to the genomic structures of bovine immune genomic loci, including MHC, IG and TR.

We updated the main text between line 83-85 in our revised manuscript, and added the references related to bovine immune diversity, including MHC, CDRH3 and $\gamma\delta$ TCR.

We added the ref (PMID: 32191811, Rosen, Bickhart et al. 2020) of ARS-UCD1.2 (line 89 in the revised manuscript).

We added ref (PMID: 19568741, Reinink and Van Rhijn 2009) to demonstrate the large expansion of TRAV genes (line 211 in the revised manuscript).

We added reference (PMID: 8968166, Arun, Breuer et al. 1996) (line 263 in the revised manuscript) to demonstrate the usage preference between IGL and IGK.

For the assembly of haplotypes of MHC, we updated the main text (line 283-287 in the revised manuscript), and cited the series studies of MHC (PMID: 26227296, Schwartz and Hammond 2015; PMID: 34802191, Schwartz, Maccari et al. 2022; PMID: 33685702, Bakshy, Heimeier et al. 2021).

We added reference (PMID: 2891610, Andersson and Rask 1988) to demonstrate the lack of DP genes in cattle (line 295 in the revised manuscript).

That would be much appreciated if the reviewer could kindly point out any areas where we may have fallen short in literature referencing.

Although the cattle genome assembly and the complete haplotypes of the MHC regions were published previously, our work reported a new assembly, NCBA1.0, represented a more complete and accurate reference of cattle genome, particularly the immune genome. Our data, in conjunction with the previous published data, can serve as the blueprint for the final gapless telomere-to-telomere cattle genome assembly in the near future.

2. The importance of immunogenetic loci, particularly germline encoded, is in the polymorphism of each alleles, alongside the structure. For example, MHC haplotypes are defined by both in tandem. Therefore, the level of detail of sequence and comparison across all the loci describes precludes publication if the focus is to remain on this level of immunogenetics.

Response: We greatly appreciate the reviewer's suggestions and agree with the reviewer to "*concentrate more on the assembly...*". Following the reviewer's suggestions, we devoted more effort and performed in-depth analysis on the genome assembly. We evaluated the correctness of NCBA1.0 assembly and demonstrated its high accuracy at base level (**detailed description in response to minor comment 2 and Figure. R6, below**). The evaluation of continuity (N50 metrics, gap numbers and contig numbers, **Fig. 1a and Supplementary Figure. 1e-f, 2c in our revised manuscript**) and completeness (mapping ratio of sequencing reads, evaluation of conserved gene sets, **Figure. 1b-d, and Supplementary Figure. 1g-h, 4-5 in our revised manuscript**) was well discussed in our original manuscript. Moreover, we built 3D genome heatmaps of NCBA1.0 assembly with *in situ* Hi-C method, and all chromosomes showed clear intra-chromosomal diagonal signals with no significant inter-chromosomal signals (**Figure.R4C, also shown as Supplementary Figure. 2a in our revised manuscript**), demonstrating the low mis-assembly rate of structural variations. Taken together, we represented a more complete and accurate reference of cattle genome.

Detailed annotation of immunogenetic regions not only provided a blueprint for cattle immune genomic loci, but also demonstrated the completeness and accuracy of our assembly. Moreover, following the suggestions by reviewer 2, we drafted a new title of our manuscript as "De novo Genome Assembly Depicts

the Immune Genomic Terrain of Cattle", which emphasizes more about the genome assembly.

We agree that conducting haplotype for immune loci is of great importance, as it can provide deeper insights into the genetic variation and diversity within these regions. To better address this question, we set up a pipeline integrated with MHC gene allele mapping and gene full-length transcripts validation (**Figure. R2A**). Utilizing the pipeline, a stringent delineation of the gene structure of MHC II haplotypes was reperformed (**Figure. R2B, also shown as Figure. 6f in our revised manuscript**).

3. Linked to the above, the sequencing error rate was mentioned for PacBio, but I didn't see it discussed at all for MinION data this is the basis of the reference used. Considering the focus on allele polymorphism and relatively low coverage this has to be a major focus of the paper, quantified and evidenced.

Response: We thank the reviewer for this important point. The quality and accuracy of ONT sequencing technology has been significantly improved through advances in both sequencing chips (R10 kit) and base-calling algorithms (guppy). We dissected the sequencing quality of our data. As shown in **Figure. R4A**, by using the "SUP" mode for base-calling, the average read quality of total raw ultra-long reads is around Phred score 14. 60.8% of the bases with quality scores above 20, indicating that a majority of the sequencing data is of high quality, with relatively low error rates. In addition to the evaluation of continuity (N50 metrics, gap numbers and contig numbers, **Fig. 1a, Supplementary Figure. 1e-f, 2c, and Supplementary Table 2,3,5,6,7 in our revised manuscript**) and completeness (mapping ratio of sequencing reads, evaluation of conserved gene sets, **Figure. 1b-d, Supplementary Figure. 1g-h, 4-5, and Supplementary Table 4,8 in our revised manuscript**) of the new assembly, we further evaluated the correctness of NCBA1.0. Illumina reads were mapped onto the assembly and homozygous variations were called. We identified 10,813 homo SNPs (error rate 0.0004%) and 10,738 homo Indels (error rate 0.0008%) with minimum coverage of 5, suggesting an average consensus single base accuracy of Phred Q51(**Figure. R4B**). We also used Pilon to identified breakpoints and no breakpoint was detected (**Figure. R4B**). The high accuracy of the assembly and allele polymorphism of immune genomic loci is further confirmed by the exact mapping of full-length transcripts

back to the gene loci (**Figure. R4C**). Moreover, we built 3D genome heatmaps of NCBA1.0 assembly with in situ Hi-C method, and all chromosomes showed clear intra-chromosomal diagonal signals with no significant inter-chromosomal signals (**Figure. R4D**), demonstrating the low mis-assembly rate of structural variations.

Thus, based on the high quality ultra-long ONT reads, we were able to finish the high-quality bovine genome assembly, which enables us to focus on the allele polymorphism, including MHC gene locus. We included the above new data in our revised manuscript (**Supplementary Figure. 1d, and Supplementary Figure. 2a, b**).

Minor comments:

4. Line 158 and Figure 2; to clarify the situation with IGH in ARS-UCD1.2, much of the IGH is incorrectly assembled to Chromosomes 20 (positions: 71881100-71974595) and 21 (positions: 1-411439) and some of it is unplaced (e.g., NKLS02001456.1). The text and figure should reflect this as it is currently misleading.

Response: We thank the reviewer for the great suggestion. We mapped the IGH genes to genome ARS-UCD1.2 and found that, as the reviewer pointed out, IGH is incorrectly assembled to three regions (**Figure. R5**). On chr20 (positions: 71881100-71974595) only IGH constant genes including IGHG, IGHE and IGHA were annotated; on chr21 (positions: 1-411439) the intact V/D/J/C gene segments were assembled, while several IGHV genes were located on the unplaced scaffold NKLS02001456.1 (**Figure. R5**). We included these new data (**Supplementary Figure. 7 in our revised manuscript**, line 192-193) and edited **Figure 2A** in our revised manuscript.

5. Lines 177-8; While novel loci could be given provisional names in the manuscript, renaming gene segments that already have established IMGT names will create confusion. It would be more useful to retain the established IMGT nomenclature than to re-invent it. IMGT nomenclature is pseudo-positional – it is not strictly required to maintain positional numbering in order to account for gene content variation. For example: PMID: 24934119

Response: We fully agree with the reviewer that renaming gene segments that already have established IMGT names may create confusion. To address this point, we made revisions accordingly.

(1) To maintain consistency with the IMGT database, gene names of IGH, IGK and TRG were retained the same with IMGT nomenclature in NCBA1.0.

(2) For newly discovered genes, provisional names with asterisk labels were assigned. For example, we updated genome annotation of two newly discovered IGHV genes (IGHV2-48* and IGHV1-49*, **Figure. R6**).

(3) We kept the same gene name from IMGT except the genomic loci containing gaps (IGL, TRA/D and TRB), and we renamed genes in newly closed gap regions based on their positional order in NCBA1.0.

We updated the gene nomenclatures in all related figures and cited reference ([PMID: 24934119](https://pubmed.ncbi.nlm.nih.gov/24934119/), Schwartz and Murtaugh 2014) in line 173 in the revised manuscript.

6. Lines 182-3 (example); there are several places in the text that suggest that the newly described assembly “corrects” the reference assembly. Care should be taken when

making these statements since the two genomes are derived from different individuals. It is quite possible (and perhaps expected) that haplotypic variation in gene content exists in these repetitive immune-related gene complexes. Similarly, the new assembly is not simply a new “version” (line 192), since it is not an update to the existing reference assembly (i.e., it is an entirely different individual!)

Response: We thank the reviewer for the important suggestions. Accordingly, we carefully revised the entire manuscript and made appropriate revisions to address any potentially misleading statements.

We rephrased the statement of “We also corrected the repeat numbers of IGLJ-IGLC clusters from nine to six” as “It is worth noting that we have characterized six repeats of IGLJ-IGLC clusters, while in ARS-UCD1.2 the number of repeats is nine” (line 196-198 in the revised manuscript). We rephrased the statement of “We corrected the tandem repeat regions within the IGH locus from three to two” as “We characterized and rigorously demonstrated the tandem repeat patterns within the IGH and IGL regions” (line 351-352 in the revised manuscript).

Besides, we also revised statement of “compared to the previous genome version, ARS-UCD1.2” to “compared to ARS-UCD1.2” in line 206 of the revised manuscript.

7. Line 273; Ref 32 describes the felid MHC. A more appropriate reference for the cattle MHC, and which established the NC6-NC10 nomenclature, is: PMID: 34802191

Response: Following the reviewer’s suggestions, we added two references (PMID: 34802191, Schwartz, Maccari et al. 2022; PMID: 26227296, Schwartz and Hammond 2015) to explain the MHC gene nomenclature (line 287 in the revised manuscript).

8. Line 270; Ref 8 is for the human MHC. A more appropriate reference for cattle (MHC I) is: PMID: 24934119. Also, the lack of DP genes in cattle was already reported long ago (PMID: 2891610)

Response: We appreciate the reviewer’s suggestions. As in our response to minor comment 5, we already cited the reference (PMID: 24934119, Schwartz and Murtaugh 2014, line 173 in the revised manuscript) that characterized a polymorphic IGLV gene in pigs to explain nomenclature for novel genes. We also added

references as the reviewer mentioned (PMID: 34802191, Schwartz, Maccari et al. 2022; PMID: 26227296, Schwartz and Hammond 2015 in line 285 in the revised manuscript; PMID: 2891610, Andersson and Rask 1988 in line 295 in the revised manuscript).

9. Line 448; “... validated by four irrelevant people.” I think “independent” is maybe intended here instead of “irrelevant”?

Response: We made emendation according to the reviewer’s suggestion.

10. Line 456; please include a citation for IPD-MHC (PMID: 2891610)

Response: Thanks for the reviewer’s valuable suggestions. As mentioned in minor comment 8, reference (PMID: 2891610, Andersson and Rask 1988, line 295 in the revised manuscript) demonstrated the lack of DP genes in cattle. We also cited reference (PMID: 27899604, Maccari, Robinson et al. 2017, line 490 in the revised manuscript) for IPD-MHC.

REVIEWERS' COMMENTS

Reviewer #1 (Remarks to the Author):

The authors have done a good job of addressing the scientific questions submitted following review of the first draft of the paper. In general I am satisfied with the changes made.

However, there are a number of small issues that still need to be addressed:

Line 43/44 – not sure that Ig/TR undergo 'maturation' and instead of 'one' segment it is more usual to refer to 'individual' segments

Line 51 – not sure use of 'alleles' is appropriate here as the allelic diversity has not be evaluated in this study

Line 273-276 – what parameter was used to quantify diversity?

Major issue is now language. Throughout the manuscript there are a substantial number of examples where there are errors in the language and awkward phrasing that detracts from the enjoyment of reading the paper. It is essential that the manuscript is subjected to a robust language review and proof-reading before it can be considered for acceptance.

Reviewer #2 (Remarks to the Author):

This revision is considerably improved by the thorough responses of the authors to the three reviews of the original manuscript. The work reported is a significant advance in the analysis of the genome of cattle. The authors carefully addressed the concerns, questions and comments provided by three reviewers of the original submission.

Issues with the writing continue to be of concern. It would be very good if the quality of the writing were improved. Yes, some of the previously noted problems were addressed, but the writing is still "rough". The meaning is generally understandable but parts of the text are such that they slow readers down from moving through the report. Given the quality of the report, it would be really good, if at all possible, to work more (with the help of an experienced writer of English) to improve the text.

For example starting on line 20, "...are the most intricate genomic regions across the the whole genome while remain poorly understood do to their huge genomic span and genetic

complexity."; needs improved easily. A possibility is "... intricate genomic regions that remain poorly understood because of their genetic complexity and size."

Starting on line 24, "advanced technologies. We provide detailed annotation of these three important immunogenomic loci."

Starting on line 25, "In contrast to the current cattle reference genome (ARS-UCD1.2), 156 gaps are closed and 467 scaffolds are located in our assembly."

It goes on from there. Generally word usage is okay but style is in need of improvement. There is one error where "bread" is used for "breed".

It is wonderful to see the improvements in the manuscript including the additional analysis and the addition of references to relevant previous work.

Given the achievement, the additional work on presentation is just a small remaining part.

Reviewer #3 (Remarks to the Author):

I commend the efforts to refocus and improve the manuscript. I think as it stands there is a good balance between genome and immunogenetic variation. I think there is clearly enough novel information of high enough quality to publish.

John Hammond

Point-by-Point Response:

Reviewer #1

1. The authors have done a good job of addressing the scientific questions submitted following review of the first draft of the paper. In general, I am satisfied with the changes made. However, there are a number of small issues that still need to be addressed:

- *Line 43/44 – not sure that Ig/TR undergo ‘maturation’ and instead of ‘one’ segment it is more usual to refer to ‘individual’ segments.*
- *Line 51 – not sure use of ‘alleles’ is appropriate here as the allelic diversity has not be evaluated in this study*

Response: We thank the reviewer for the encouraging comments. For line 43/44, we rephrased the sentence from "Ig/TR undergo maturation" to "**B/T cells undergo maturation**" and revised "one segment" as "**individual segment**" according to the reviewer's suggestion.

For line 51, "alleles" is now replaced with "regions" for better clarity and accuracy.

- *Line 273-276 – what parameter was used to quantify diversity?*

Response: By analyzing the V(D)J recombination profiles, we determined the preference and frequency of V-D-J segment usage during the recombination of TR and IG. As shown in Figure 5d-j, we described the ‘diversity’ according to the numbers of types of V(D)J recombination events. We did not use any quantitative method to compare the diversity. We apologize for our unclear description, and we revised the text accordingly (Line 267-269 in our revised manuscript).

2. Major issue is now language. Throughout the manuscript there are a substantial number of examples where there are errors in the language and awkward phrasing that detracts from the enjoyment of reading the paper. It is essential that the manuscript is subjected to a robust language review and proof-reading before it can be considered for acceptance.

Response: Following the reviewer's suggestions, we polished the text with the help of an experienced writer of English and formatted the manuscript according to the editorial requests. We highlighted all changes in the text file.

Reviewer #2

1. This revision is considerably improved by the thorough responses of the authors to the three reviews of the original manuscript. The work reported is a significant advance in the analysis of the genome of cattle. The authors carefully addressed the concerns, questions and comments provided by three reviewers of the original submission.

Issues with the writing continue to be of concern. It would be very good if the quality of the writing were improved. Yes, some of the previously noted problems were addressed, but the writing is still "rough". The meaning is generally understandable but parts of the text are such that they slow readers down from moving through the report. Given the quality of the report, it would be really good, if at all possible, to work more (with the help of an experienced writer of English) to improve the text.

For example starting on line 20, "...are the most intricate genomic regions across the whole genome while remain poorly understood do to their huge genomic span and genetic complexity."; needs improved easily. A possibility is "... intricate genomic regions that remain poorly understood because of their genetic complexity and size."

Starting on line 24, "advanced technologies. We provide detailed annotation of these three important immunogenomic loci."

Starting on line 25, "In contrast to the current cattle reference genome (ARS-UCD1.2), 156 gaps are closed and 467 scaffolds are located in our assembly.

It goes on from there. Generally word usage is okay but style is in need of improvement There is one error where "bread" is used for "breed".

It is wonderful to see the improvements in the manuscript including the additional analysis and the addition of references to relevant previous work.

Given the achievement, the additional work on presentation is just a small remaining part.

Response: We thank the reviewer for the encouraging comments and greatly appreciate the reviewer's help in polishing the language of our manuscript. We rephrased the sentences according to the reviewer's suggestions (listed above). Further, we revised our manuscript with the assistance of an experienced English writer and formatted the manuscript according to the editorial guidelines. We highlighted all changes in the text file.

Reviewer #3

1. I commend the efforts to refocus and improve the manuscript. I think as it stands there is a good balance between genome and immunogenetic variation. I think there is clearly enough novel information of high enough quality to publish.

Response: We greatly appreciate the reviewer's encouraging comments.